# COMMON CORPUS: THE LARGEST COLLECTION OF ETHICAL DATA FOR LLM PRE-TRAINING

**Pierre-Carl Langlais**[*]   **Pavel Chizhov**[*†]   **Catherine Arnett**[*‡]   **Carlos Rosas Hinostroza**[*]

**Mattia Nee**[*]        **Eliot Krzystof Jones**[*]        **Irène Girard**[*]        **David Mach**[*]

**Anastasia Stasenko**[*]        **Ivan P. Yamshchikov**[*†]

Correspondence: `pierre-carl@pleias.fr`

## ABSTRACT

Large Language Models (LLMs) are pre-trained on large amounts of data from different sources and domains. Such datasets often contain trillions of tokens, including large portions of copyrighted or proprietary content, which raises questions about the legal use of such models. This underscores the need for truly open pre-training data that complies with data security regulations. In this paper, we introduce Common Corpus, the largest open dataset for LLM pre-training. The data assembled in Common Corpus are either uncopyrighted or under open licenses, totaling about two trillion tokens. The dataset contains a wide variety of languages, ranging from the high-resource European languages to some low-resource languages rarely represented in pre-training datasets. In addition, it includes a large amount of code data. The diversity of data sources in terms of covered domains and time periods opens up the paths for both research and entrepreneurial needs across diverse areas of knowledge. In this paper, we present the detailed provenance of data assembling and the details of dataset filtering and curation. We train two small language models on Common Corpus and find that they perform comparably to other models of their size, indicating that our dataset is suitable for multilingual pretraining. Common Corpus represents a key contribution to the ecosystem for open science research on Large Language Models.

🤗 `PleIAs/common_corpus`

## 1 INTRODUCTION

Large Language Models demand large amounts of training data. GPT-3 (Brown et al., 2020) is considered the first "large" language model. Trained on 300 billion tokens, GPT-3 introduced a standard training data pipeline shared by nearly all language models to date: large-scale processing of web datasets (45 TB of compressed source data from Common Crawl) and additional digitized sources (Books3). Until 2025, LLM training data sizes have grown logarithmically. The latest generation of publicly documented language models including DeepSeek v3 (Liu et al., 2025), Gemma 3 (Kamath et al., 2025), Llama 4 (Meta, 2025) or Qwen 3 (Yang et al., 2025) have been trained on 14-36 trillion tokens. Even "small language models" (Wang et al., 2025) rely on large amounts of training data to fit scaling laws: Qwen 3 0.6B was trained on 36 trillion tokens, which is 3,000 times more than Chinchilla-optimal estimates (Hoffmann et al., 2022).

As data quality has increasingly been an area of concern, the collection, maintenance, processing, and filtering of data has become one of the main costs in language model training. But data curation

---

[*]PleIAs, Paris, France 🔗 https://pleias.fr/

[†]CAIRO, Technical University of Applied Sciences Würzburg-Schweinfurt, Germany

[‡]Work done at PleIAs. Now at EleutherAI.

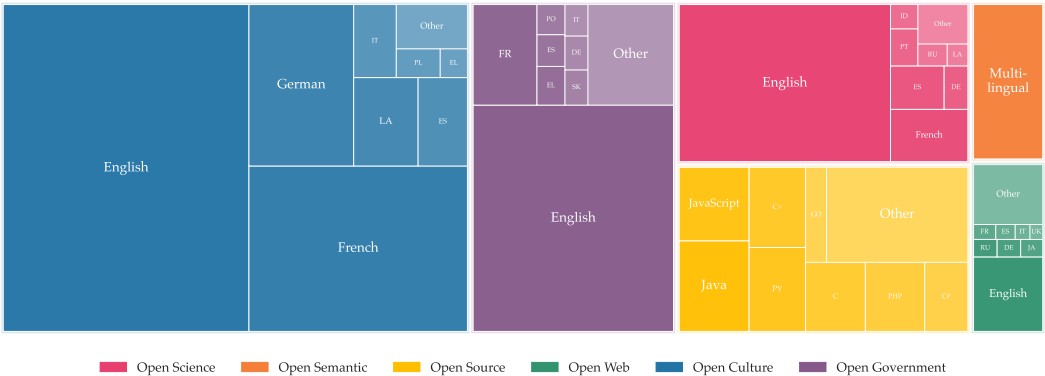

Figure 1: Proportional treemap of Common Corpus collections and their most popular languages.

at scale may also have hidden costs: negative externalities affecting competing markets, the digital commons, and society at large. While data scraped from the web is publicly available, it is not always in the public domain. Most web data lacks sufficient metadata to determine whether it is openly licensed. NLP practitioners have relied on the protection of fair use, claiming that the transformative nature of the use of the data allows them to leverage this data to train language models. There are increasingly more legal challenges to the use of this data. The New York Times sued OpenAI for copyright infringement, alleging that OpenAI trained their models on NYT articles (Roth, 2023; Pope, 2024). Due to concerns about indirect commercial exploitation, many rightholders have implemented either hard technical measures or legal provisions against model training. In 2024, it was estimated that for Terms of Service crawling restrictions, a full 45% of C4 is now restricted (Longpre et al., 2024b) and 5% is fully blocked for scraping with a disproportionate impact on quality sources (Longpre et al., 2024b). Restrictions not only affect LLM pre-training but also the quality of search engine indexation and a variety of research projects that analyze and collect content at scale. Even projects dedicated to knowledge access have faced significant pressure from AI crawlers and implemented protections that negatively impact access and user experience.

Legal uncertainties have significantly impeded the development of open science research on LLMs. Previously reproducible research artifacts have been removed or taken down, impacting pre-training data, continuous pre-trained models, and evaluation datasets. Books3, which has been used to train numerous models, faced legal challenges (Brittain, 2023). The original dataset was ultimately removed due to a DMCA take-down (Van der Sar, 2023). The LAION dataset was demonstrated to contain CSAM (Birhane et al., 2021; Thiel, 2023), and taken down (LAION, 2023), and then re-released once suspected CSAM was removed (LAION, 2024). The Dutch model GEITje was taken down (Rijgersberg, 2025), due to complaints about training on the Dutch Gigacorpus, in order to avoid legal disputes. Finally, the widely used benchmark, the Mathematics Aptitude Test of Heuristics (MATH) dataset (Hendrycks et al., 2021), was removed from Hugging Face via a DMCA takedown. All of these artifacts, which were released to further open development and evaluation of language models, were removed suddenly, making previous work unreplicable. These takedowns and legal challenges also represent a sizeable loss of investment for developers, who are often independent or small research organizations.

In part as a reaction to the use of publicly available but not permissively licensed data, web text is also becoming harder to acquire and use. In an analysis of popular datasets such as C4 (Raffel et al., 2020), RefinedWeb (Penedo et al., 2023), and Dolma (Soldaini et al., 2024), Longpre et al. (2024b) found that just in the last year, 5% of all tokens in C4 now have restricted use, with a disproportionate number of those tokens coming from the best-maintained, most critical sources. This is largely due to changes in content owners' and hosts' preferences, which are changing to no longer allow scraping, especially for the purposes of training AI models.

Since 2024, several initiatives have emerged to collect open data in English with clear licensing. This includes: C4C (Habernal et al., 2016); Open License Corpus (Min et al., 2024), a 228 billion token corpus from a mix of public domain texts and open source code under free licenses; KL3M (Bommarito et al., 2025), a 1.2 trillion tokens corpus of administrative texts and structured data

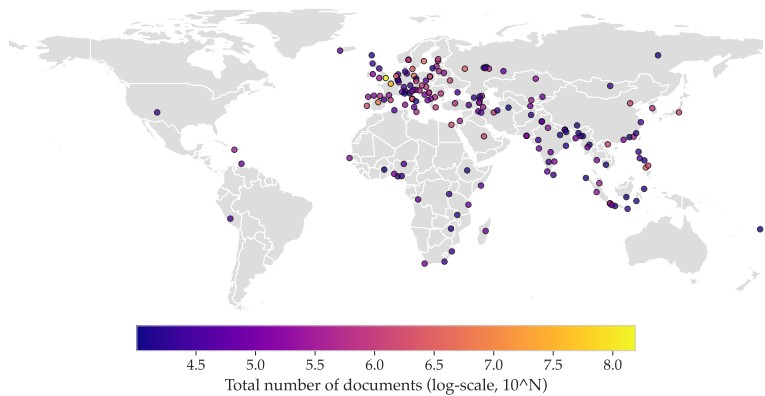

Figure 2: A schematic world map of languages in Common Corpus with a log-scaled distribution of document counts. For each language, we chose a city that is located in the region where this language is most specific to. To avoid outliers, we show only languages with 10,000+ documents.

mostly from the US federal public domain; Common Pile (Kandpal et al., 2025), a data collection of 1 trillion tokens from a variety of recent sources, including a filtered common crawl (Creative Commons Common Crawl). All these projects are monolingual, restricting in effect the reach of language models to the English-speaking audience. In contrast, the most ambitious multilingual collection of permissive content predates Large Language Models: C4C contains 12 million web pages in more than 50 languages filtered by Creative Commons Licenses (Habernal et al., 2016).

Common Corpus has grown to become the largest fully open pre-training dataset at about **2 trillion tokens** and the only one in its size range having high multilingual diversity. Through this release, we show that open LLM research and development is possible while meeting legal and regulatory requirements — in compliance with even the strictest AI regulations, such as in the European Union. In this paper, we detail the composition of Common Corpus and the process of data collection and curation, and license clearing. Despite its size, Common Corpus is still far from covering the entire range of available resources. We attribute this discrepancy to an *open data paradox* as major sources of open content are paradoxically little visible online and even more so in the leading pre-training sources. By describing the unique challenges coming with the aggregation of large open source, we aim to inspire further initiatives. We also train two small language models on our dataset and find that it offers comparable performance to existing multilingual models.

## 2 ABOUT COMMON CORPUS

When talking about Common Corpus data, we use the word **"open"** in the strongest sense. Not only is the data available, but we also provide essential details about the data provenance, data processing, and important information about the contents of each dataset. The Open Source Initiative has also defined open-source AI in terms of openness of use, where open means that use is permitted for "any purpose and without having to ask for permission" (Open Source Initiative, 2024). To achieve this, models must be trained on datasets that are free from copyright or other legal limitations. This is currently a limitation of existing open datasets for training LLMs.

Common Corpus, therefore, provides valuable training tokens that will not be subject to the same restrictions. Additionally, the data in Common Corpus are different from other corpora, primarily composed of web text. It contains multilingual data in a variety of high- and low-resource languages (see Figure 2 for language distribution), covering diverse genres, time periods, and domains (in Section 4, we detail each part of the dataset). Therefore, it contributes to data diversity in the open pre-training data ecosystem. This is important for developing powerful and generalizable model performance. Common Corpus can be used on its own or in conjunction with existing open datasets, according to one's needs and the desired use case of a language model.

Common Corpus was developed with consideration for developing best practices for open-source LLM development (The AI Alliance, 2024; Longpre et al., 2024a; Duprieu & Berkouk, 2024; Baack et al., 2025). We highlight our adherence to those suggested by Baack et al. (2025):

- **Provide useful documentation.** We provide information about dataset provenance and processing (Sections 4 and 5) and share key statistics to help potential users understand the applications of the dataset. Dataset documentation improves reproducibility, helps prevent misuse, and aids downstream users to best utilize the dataset (Longpre et al., 2024a).
- **Follow and record preference signals.** In the metadata, we include the source URL and license information for the vast majority of the corpus.
- **Increase diversity and involve local communities to identify relevant data sources.** This dataset includes data from a variety of languages, coming from high-quality sources, and which was never machine-translated.
- **Share advancements to foster reciprocity and give back.** In addition to the dataset, we release many of the tools we developed in order to create the final dataset (Section 5).
- **Do not use openly licensed data without regard for its quality or fitness for purpose.** In particular, for the dataset in the public domain, we engage in extensive OCR correction and toxicity filtering in order to bring datasets up to standard (Section 5).
- **Do not capture highly sensitive data.** We remove personally identifiable information from our datasets (Section 5).

Common Corpus aims to support the pre-training of fully open and auditable LLMs by making it legal to release the source even without the provision of fair use. It has been used to create a wider range of language model artifacts, including multimodal datasets, classifiers, synthetic datasets, and benchmarks. Beyond the main dataset, Common Corpus works as an open science infrastructure dedicated to the entire lifecycle of language models. As defined by UNESCO, it is a shared research infrastructure that is needed to support open science and serve the needs of different communities (Unesco, 2021). We argue this is the first point in time where there has been sufficient knowledge and infrastructure to collect and clean a dataset on this scale, which meets the legal and ethical criteria we have outlined.

Common Corpus is composed of six collections: Open Government, Open Culture, Open Science, Open Web, Open Code, and Open Semantic (Figure 1), and in total amounts to **1,998,647,168,282** tokens[1]. The token counts in each collection are listed in Appendix B. To highlight the diversity, we visualize the timeline of the collected documents and embeddings of a subsample in Figure 3. Each collection comprises multiple datasets, for which we provide details on provenance and other key information in the corresponding subsections. Each data object contains a license, language(s), a collection or domain of specialization, and other metadata, allowing one to filter out a desired subset. We note that the majority of the data in Common Corpus is in the public domain (see Appendix B).

Common Corpus is multilingual (Figure 2 and language distribution in Appendix B), with at least 10B tokens for nine languages[2]. The issues discussed above for making datasets are compounded in languages other than English. Even in relatively high-resource languages like French, these problems are compounded by the fact that there is much less data available, and most tools generalize poorly to languages other than English. Additionally, Kreutzer et al. (2022) showed that many multilingual datasets contain a lot of low-quality or entirely unusable data. Many of the existing datasets they analyzed contained less than 50% of usable text, with 15 sources containing no usable data at all.

## 3 RELATED WORK

Table 1 illustrates the similarities and differences between prominent open pre-training datasets and Common Corpus. Multiple other efforts, including C4 (Raffel et al., 2020), ROOTS (Laurençon et al., 2022), DCAD-2000 (Shen et al., 2025), and FineWeb 2 (Penedo et al., 2025), contain multilingual data from several distinct domains, but consist mostly of web crawl data with not only permissive

---

[1]The token counts are calculated using PleIAs tokenizer trained on a representative subsample of Common Corpus, available at: 🤗 `PleIAs/Pleias-350m-Preview`

[2]Language distribution was computed using the fastText language identification model.

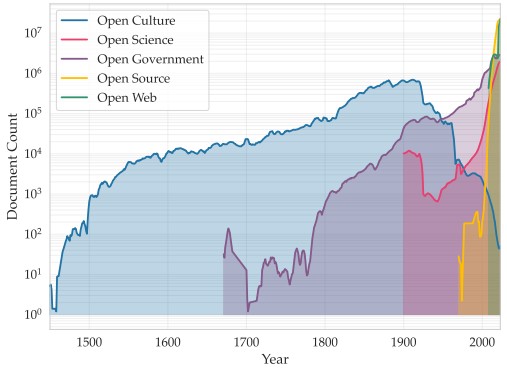 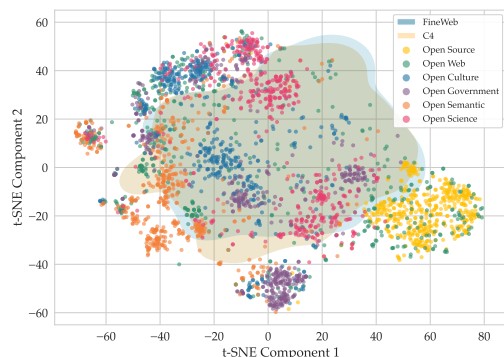

(a) A timeline of the main collections with their numbers of documents in the Common Corpus.

(b) A two-component t-SNE visualization of subsets of Common Corpus collections, C4, and FineWeb.

Figure 3: Temporal and semantic overview of the Common Corpus collections.

licenses. Dolma (Soldaini et al., 2024) expands beyond web crawl, but is primarily comprised of English data. KL3M (Bommarito et al., 2025) is offering strictly permissive licensing beyond crawled data, but it is limited to administrative and legal documents in English. Common Pile (Kandpal et al., 2025) shares Common Corpus's commitment to varied data with only open licensing, but is English-only. Common Corpus is unique in satisfying all four criteria simultaneously: multilingual and multi-domain coverage, data sources beyond web crawls, and fully open licensing.

|                  | KL3M | Dolma | C4 | ROOTS | DCAD 2000 | FineWeb 2 | Common Pile | Common Corpus |
|------------------|------|-------|----|-------|-----------|-----------|-------------|---------------|
| Multidomain      | ✗    | ✓     | ✓  | ✓     | ✓         | ✓         | ✓           | ✓             |
| Beyond Web Crawl | ✓    | ✓     | ✗  | ✗     | ✗         | ✗         | ✓           | ✓             |
| Multilingual     | ✗    | ✗     | ✓  | ✓     | ✓         | ✓         | ✗           | ✓             |
| Open data        | ✓    | ✗     | ✗  | ✗     | ✗         | ✗         | ✓           | ✓             |

Table 1: Comparison of the contemporary datasets for LLM training.

We also stress the difference in data sources. An investigation of the top 1000 domains of FineWeb shows very low overlap with the Common Corpus: less than 2% of pages and 1% of domain names. Within overlapping collections, content can differ substantially: we focused on large PDF sets, whereas agnostic crawl pipelines mostly retrieve HTML pages (e.g., abstracts from scientific papers). This means that Common Corpus mostly adds new, diverse content to the open pretraining ecosystem and significantly differs from crawl corpora, as we also illustrate in Figure 3b.

## 4 PROVENANCE

In this section, we present the details about collections that comprise the Common Corpus, accompanied by the information about the data sources and the main included languages in Appendix D.

### 4.1 OPEN GOVERNMENT

Open Government is a set of financial, legal, and administrative data in the public domain. In total, the dataset contains more than 407B tokens and comprises two main datasets: Finance Commons and Legal Commons. See Appendix D.1 for detailed data composition.

**Finance Commons.** This is the largest collection of financial documents in the public domain, comprising more than 14 billion words (more than 23 billion tokens). The documents come from a wide time range, all the way to 2024. Like many of our other datasets, Finance Commons is also multilingual. Most of the documents are in English, French, and German, but there are also texts in other languages such as Romanian, Bulgarian, and Latvian. Additionally, this is a multimodal dataset. It includes more than 1.36 million original PDF documents from AMF and the WTO. The documents

constitute a wide coverage of in-house layouts and formats produced by industrial and economic sectors. This makes this dataset ideal for developing the next generation of open-data multimodal models. One application for this dataset is to develop vision-language models (VLMs) for advanced document segmentation and processing. These documents also contain vast amounts of structured data, which is also a promising area of research that Finance Commons can help drive forward.

**Legal Commons.** This is a collection of legal and administrative datasets. The datasets come mostly from the EU and the US and cover a wide range of languages. These datasets are useful for developing language models with legal knowledge, as well as models that are ideal for document processing in official administrative applications.

## 4.2 OPEN CULTURE

Open Culture is an aggregation of vast cultural heritage datasets containing both monographs and periodicals for over 13 languages: French, English, German, Spanish, Portuguese, Italian, Dutch, Luxembourgish, Danish, Swedish, Serbian, Czech, and Greek. There are also small portions of data in other languages, such as Arabic, Bengali, Latin, Persian, Russian, Sanskrit, and Urdu.

**Composition.** A large part of Open Culture is compiled from Collections As Data (CAD) — large dumps of texts, datasets, PDFs, and even raw XML output (METS/ALTO). CAD initiatives considerably simplify dataset aggregation and are a major contribution to the digital commons ecosystem. The remaining sources have been collected on a resource-by-resource basis using APIs and other standard retrieval methods whenever available. The largest extractions of this kind include Internet Archive (about 2 million monographs) and Delpher (50,000 Dutch monographs and periodicals filtered to match the Dutch copyright law for public domain). We managed to compile a large multilingual collection despite such challenges, such as poor OCR quality, through the development of OCR correction tools (see Section 5), text segmentation issues, and sometimes irrecoverable deterioration of the original support. For the detailed dataset composition, refer to Appendix D.2.

**Licenses.** All Open Culture documents are in the public domain, which means their copyright has expired after a given term, and there are no limitations on their reuse. For certain content, or in cases where we could not rely on the guarantee of established cultural heritage institutions, we implemented our own internal rights verification process. This process follows specific criteria, including author life and data object creation time, and takes into account that we only collected cultural heritage content from institutions based in the US or the EU (see the complete criteria list in Appendix E).

**Value.** Open Culture data is also rich from a cultural and stylistic standpoint and can be used to train multilingual language models with more diverse and creative writing styles. As LLMs are trained on extremely large corpora to maximize next-word prediction accuracy, LLM-generated text can often lack in personality and be boring or generic (Jones & Bergen, 2024). This feature of language models stands in contrast with one of their most common uses. In an analysis of WildChat (Zhao et al., 2024), a dataset of 1 million user interactions with ChatGPT, Longpre et al. (2024b) found that over 30% of user requests involved creative compositions such as fictional stories, role-play, or poetry generation. At the same time, creative writing is poorly represented among datasets used to train LLMs, which mainly comprise web text (Longpre et al., 2024b). Therefore, Open Culture contributes data that can be used to train models for creative writing without violating copyright law. In addition, as many of the Open Culture datasets are historical (coming from the 18th-19th centuries, or even earlier; see Figure 3a), this collection also enables the development of historical language models. The metadata includes the document creation year, which enables researchers to develop language models with a cutoff of the training data creation date.

## 4.3 OPEN SCIENCE

The Open Science collection includes scientific papers and other documents (theses, book reviews, clinical trials, etc). Following the development of a global open access movement, these documents have been made increasingly available in open archives (preprints) or directly through open science publishers and infrastructure. Scientific content has become a primary focus of training data, due to its impact on reasoning capacities. Yet, the lack of licensing information has until now partly hindered reuse. The Semantic Scholar Open Research Corpus from Allen AI includes 81.1 million

articles in English under an Open Data Commons Attribution License, allowing for the free reuse of the aggregated metadata while still acknowledging the remaining copyright of individual authors (Lo et al., 2020). The Pile incorporated data from arXiv and PubMed Central, also exclusively in English (Biderman et al., 2022). Finally, the BigScience project assembled several curated multilingual scientific datasets like the French HAL as part of the training data for Bloom (Scao et al., 2023).

The Open Science collection was made possible largely due to the recent development of OpenAlex (Priem et al., 2022), the largest open catalogue of scientific documents. OpenAlex maintains an expansive API search engine tracking detailed metadata for each indexed item, including the licensing, as well as a link to the original resource, which is generally in PDF format. We filtered OpenAlex on the following three licenses: CC-BY, Public Domain/CC0, and CC-BY-SA. The largest share of resources is available under CC-BY, which is currently the recommended license by the Open Access definition. Open Science also includes smaller subsets, such as a direct extraction of arXiv articles available in CC-BY and some European-specific resources not currently well indexed on OpenAlex (the exact distribution of token counts can be found in Appendix D.3).

Due to the specificity of open scientific publishing, the Open Science collection has less linguistic diversity, with nearly 85% of documents currently available in English.

## 4.4 OPEN CODE

The Open Code collection comprises code data under a wide variety of free licenses, which allows NLP practitioners to train models on public domain code for either coding applications or in order to improve certain model performance on natural language reasoning, world knowledge tasks, mathematics, and structured output tasks (Aryabumi et al., 2025; Petty et al., 2024; Ma et al., 2024). The code data we use comes from the Stack v1 and v2 (Kocetkov et al., 2023; Lozhkov et al., 2024). The Stack v1 contains 6.4TB of data and covers 30 programming languages, while the Stack v2 is approximately ten times bigger at 67.5TB and covers over 600 programming languages. All the code data is made available with a direct link to the original resource on GitHub. In total, Open Code contains 283,227,402,898 tokens (see most common languages in Appendix D.4).

To prepare the collection, we ran a pipeline of varied filters. We first removed files that were not in our desired set of languages and formats according to their file extensions, including SVG files containing mostly encoded shapes, data storage formats: `csv`, `json`, `json5`, `jsonld`, and other file types with non-informative content, typically in small amounts: `python-traceback`, `unity3d-asset`, `numpy`, and `http`. We then filtered out the licenses to keep only permissive ones. To discard the low-quality data, we ran a series of manual filters described by Lozhkov et al. (2024). In addition to those, we removed files consisting of 75% or more of digits, which are mostly files containing raw numeric data. Before the filters, we also replaced sequences of `[\r]+\n` with `\n` and recalculated line lengths to avoid false positives by maximum line length.

## 4.5 OPEN WEB

In accordance with the general focus of Common Corpus on curated content, the Open Web collection currently includes four major web sources:

**Wikipedia** and **Wikisource.** Wikimedia projects are popular sources for language model training data due to their reliability, extensive coverage, and textbook-like style. Despite this centrality, there is still a range of unresolved challenges with the most common versions available for training. The raw source of Wikimedia projects is made available in a specific *mediawiki* syntax, including a lot of project-specific models, tags, and conventions. The parsing of models is especially not straightforward, as they can either format existing text or remove or include external content (transclusion). As part of Wikimedia Enterprise, the Wikimedia Foundation created entirely new dumps from the rendered HTML sources, which in effect ensure that they include all the text made available to readers.

**Youtube Commons**[3]**.** For YouTube Commons, we collected audio transcripts of 2,063,066 videos uploaded on YouTube under a standardized CC-BY license.

**StackExchange.** This is a collection of user-generated forums and Q&A made available under the CC-BY-SA license. We reused the version from The Pile (Biderman et al., 2022).

---

[3] 🤗 `PleIAs/YouTube-Commons`

A major aim for the future work will be the integration of web archives filtered by permissive licenses. Since 2016, several projects have attempted to reidentify Creative Commons licenses from web archives at scale, including C4C (multilingual; Habernal et al., 2016), CCCC (English; Dodge et al., 2021), and most recently multilingual Common Crawl Creative Commons Corpus (C5; Vanroy, 2025). All these projects struggled with license identification. While license mentions are frequently normalized with a direct link or logo to Creative Commons, there is no guarantee they really concern the entire content: "a blog page contains many photos, and each photo is licensed under a different CC-license type, or a blog home page with many articles, and each article is licensed under a different CC-license type." (Habernal et al., 2016). We hope this limitation could be overcome by a combination of web domain curation and fine-grained curation and annotation by a language model.

### 4.6 OPEN SEMANTIC

Semantic data is the latest set added to Common Corpus and currently includes only one collection: Wikidata. First created in 2011, Wikidata hosts 100 million documented items and several billion factual statements encoded as RDF triples. It has grown to become a critical web infrastructure, used by Google for search disambiguation and currently embodying Tim Berners-Lee's ambitious vision for "a web of data". Despite the rising interest in mixed LLM/knowledge graph methods, Wikidata has hardly been used in language models. The largest initiative to date is Kelm, a collection of 15 million synthetic sentences generated by Google from English-speaking statements (Agarwal et al., 2021). A persistent challenge has been the exclusive availability of Wikidata dumps under formats optimized for data exchange rather than language model training.

Thanks to a collaboration with Wikimedia Deutschland, the entire set of Wikidata has been adapted in natural language and added to Common Corpus. This is to date the only available textual collection of Wikidata covering the entire range of 300 languages. Data processing involved the translation of items and properties into formal language sequences as simple natural language sequences, without textual synthesis: "Q41309 — P:27 — Q171150" becoming "Franz Liszt country of citizenship Kingdom of Hungary." Within each entry, we provide all the available translations as consecutive blocks separated by a newline, anticipating that this may contribute to language alignment.

## 5 CLEANING AND CURATION

In order to curate our dataset, we developed a number of custom tools[4] to handle the issues unique to multilingual, historical, and OCRed data. We document the tools in detail in Appendix F.

**Text Segmentation.** We developed **Segmentext**, a specialized language model for text segmentation (see example in Appendix F.1). Segmentext has been trained to be resilient to broken and unstructured texts with digitization artifacts and ill-recognized layout formats. Given the diversity of the training data, Segmentext should work correctly on diverse document formats in the main European languages.

**OCR Error Detection** We developed two pipelines to determine the digitization quality of different datasets in order to determine the appropriate treatment before datasets can be used for pre-training. The first is **OCRoscope**. This is a tool based on the language identification tool `cld2` (Al-Rfou, 2013), which identifies documents with non-recognized 7-grams to provide a measure of OCR quality for at least 80 languages (for further details, see Appendix F.2). The other is **OCRerrcr**. This is a DeBERTa-v2-style language model with 400M parameters, which is trained to label OCR errors. This enables error rate estimation and supports OCR error correction. OCRoscope tends to have lower accuracy, especially for documents with fewer OCR errors. However, it is faster and less computationally expensive, which allows it to be better used at larger scales. OCRerrcr, on the other hand, achieves higher accuracy, but is more computationally intensive.

**OCR Correction.** We developed **OCRonos** model based on Llama 3 8B (Grattafiori et al., 2024). It supports the correction of OCR errors, cutting or merging of the wrong word, and overall broken text structures. The training data includes a highly diverse set of OCRed texts in multiple languages, mostly from uncorrected versions of Open Culture and Open Government. On highly deteriorated content, OCRonos can act as a synthetic rewriting tool rather than a strict correction tool (e.g.,

---

[4] All described tools are publicly released: 🤗 `PleIAs/Segmentext` ⬤ `PleIAs/OCRoscope` 🤗 `PleIAs/OCRerrcr` 🤗 `PleIAs/OCRonos` 🤗 `PleIAs/celadon`

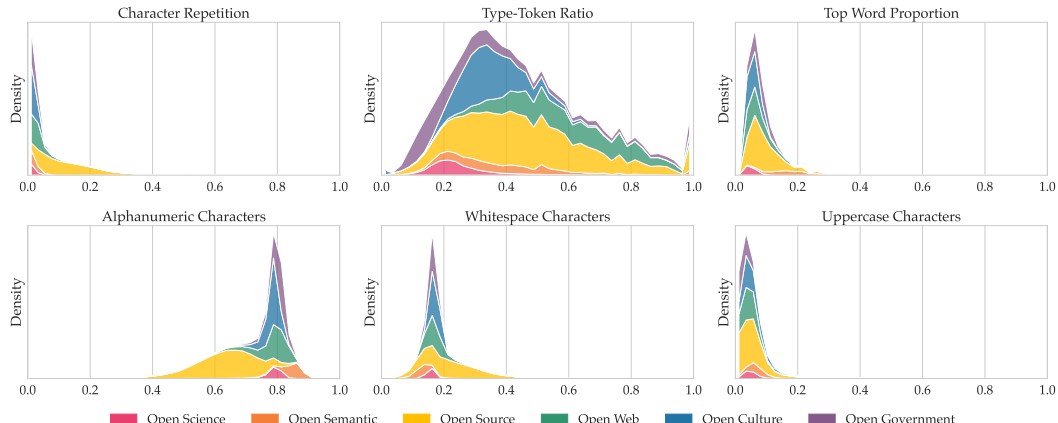

Figure 4: Stacked histograms of probability density for qualitative evaluations of Common Corpus on a sample of 300,000 documents. Metric descriptions can be found in Appendix G.

Appendix F.4). OCRonos is generally faithful to the original material, provides sensible restitution of deteriorated text, and will rarely rewrite correct words. However, when original PDF sources are too damaged for accurate OCR, unavailable, or otherwise difficult to retrieve, OCRonos provides a practical alternative for recovering usable text.

**PII Removal.** Personally Identifiable Information (PII), i.e., any information that can be used to distinguish or trace an individual's identity, is protected under legislation such as GDPR. Consequently, the new regulations put restrictions on LLM training data. In large open datasets, there is a staggering amount of personal data in widely used datasets, e.g., large quantities of phone numbers in RedPajama, email addresses in S2ORC and peS2o, and IP addresses in the Stack (Elazar et al., 2024). To identify and replace PII, we use Microsoft's Presidio (Mendels et al., 2018), an open-source state-of-the-art tool. With Presidio, we filtered out phone numbers, email addresses, IBANs, IP addresses, and URLs. With the base settings, Presidio identified on average 55-60% of texts that included phone numbers due to different possible number formats. By applying custom regular expression patterns that include most phone numbers, we increased this accuracy to 85%. Typical methods of handling PII include removing it, replacing it with tags, and partial anonymization. These transformations substantially alter the format of PII, which could undermine the model's understanding of the text or interfere with its ability to process text with real PII. Instead, we replace PII with fictitious but realistic values.

**Deduplication.** Our early experiments showed a negligible rate of duplication, which we attribute to the initial data curation: large institutions are incentivized to avoid re-digitizing the same texts. We also filtered out duplicates based on PDF metadata and used deduplicated sources wherever possible.

**Toxicity Detection.** In addition to posing legal and regulatory issues, web data is a major source of harmful and biased content (Common Crawl was shown to contain sexual content, hate speech, and racial and gender biases (Luccioni & Viviano, 2021)) and often suffers from low-quality and machine-generated text (Dodge et al., 2021). Public Domain data, such as that in Open Culture, comprises historical periodicals and monographs from at least 80 years ago. As cultural norms have changed dramatically, many of these texts do not meet modern ethical standards. Training language models on these texts would lead to the reproduction and circulation of harmful language. To address this, we developed a pipeline to filter the public domain training data. We created a multilingual toxicity classifier, **Celadon**, a DeBERTa-v3-small model (~140M parameters), trained from scratch on 2M synthetically annotated samples. Celadon and the training dataset were released as parts of a separate work (Arnett et al., 2024).

## 6 EVALUATION

We evaluate a sample from Common Corpus using various qualitative metrics and present the distributions in Figure 4 and provide details on metrics in Appendix G. Most data objects present high peaks at similar proportions in the expected range. Specific collections exhibit expected deviations:

code data has a generally higher proportion of repetitions and a lower proportion of whitespace characters due to strict syntax and the presence of punctuation, while Open Government data presents lower linguistic diversity due to fixed terminology and expressions.

We train two models on Common Corpus: PleIAs 350M and PleIAs 1.2B[5]. The architecture is based on Llama. We train a custom Llama-style tokenizer with a vocabulary size of 65536 on a representative subset of Common Corpus. PleIAs 350M is trained on a filtered subset of Common Corpus, amounting to approximately 1T tokens. PleIAs 1.2B is trained on Common Corpus and three epochs of the filtered subset. The models were trained for 2944 and 23040 H100 hours, respectively.

We evaluate our models on MultiBLiMP (Jumelet et al., 2025), XStoryCloze (Lin et al., 2022), and XCOPA (Ponti et al., 2020; see Table 2 for aggregated scores and Appendix G for per-language details). All evaluations were run using the LM Evaluation Harness (Biderman et al., 2024). Our models perform comparably to models trained on closed or non-permissively licensed data, and show outstanding performance on MultiBLIMP, which has more languages compared to other benchmarks. This is especially notable for our 350M model, which we compare to bigger models; it also outperforms models from the 1B range, except for Gemma 3 1B. Our models stably outperform OLMo 1B, which was also pre-trained on a publicly released dataset.

| Model | Ours | Gemma 3 | XGLM | BLOOM | Ours | Gemma 3 | XGLM | OLMo |
|---|---|---|---|---|---|---|---|---|
| | 350M | 270M | 564M | 560M | 1.2B | 1B | 1.7B | 1B |
| MultiBLiMP | 0.774 | 0.762 | 0.711 | 0.683 | 0.797 | 0.799 | 0.710 | 0.699 |
| XStoryCloze | 0.509 | 0.533 | 0.537 | 0.532 | 0.526 | 0.594 | 0.569 | 0.517 |
| XCOPA | 0.533 | 0.544 | 0.550 | 0.541 | 0.541 | 0.593 | 0.574 | 0.518 |

Table 2: Benchmarking results. "Ours" refers to PleIAs models pre-trained on Common Corpus.

## 7 CONCLUSION

Through the release of Common Corpus and this paper with thorough documentation of data collection and curation, we show that LLM development is possible while strictly adhering to the regulatory norms. While Common Corpus is only large enough to train small models currently, the tools and methods we used to identify and curate the data may be used to expand the amount of permissively licensed open data. We hope that Common Corpus will grow as a critical infrastructure for open science LLM research and development and inspire future initiatives in the open.

## LIMITATIONS

Common Corpus is far from collecting the whole range of available open data, which we described as the open data paradox. Therefore, the future collection of permissive data is highly encouraged by this work. Furthermore, the collected amount of data (2 trillion tokens), when used alone, as our own small language model family (see Section 6), is suitable for pre-training of models of limited size, while larger ones require significantly larger amounts of data. In addition, Common Corpus naturally does not contain data for instruction-tuning and any forms of specialized tasks. Therefore, it is not directly suitable for task-specific fine-tuning. However, due to the multilingual, temporal, and semantic diversity of data, Common Corpus opens the opportunities for the creation of ethical fine-tuning datasets.

In Section 5, we described the tools we used for the data curation, filtering, and editing. Although we used these methods responsibly and mitigated many issues overlooked by the counterparts (e.g., with toxicity detection), none of the curation methods could naturally facilitate a hundred percent accuracy. However, some issues, like OCR errors, present considerable challenges to the models and might even account for better handling of typos in the future. We would also like to mention that each data object is accompanied by sufficient metadata, and, if desired, LLM practitioners are free to filter out collections that might contain potential issues (as described in Section 5).

---

[5]Repositories: 🤗 `PleIAs/Pleias-350m-Preview` 🤗 `PleIAs/Pleias-1.2b-Preview`

ACKNOWLEDGEMENTS

Common Corpus is part of the Current AI initiative to accelerate the development of the building blocks of open AI — notably data — that serves and is shaped by the public interest. It was built up with the support and concerted efforts of AI Alliance, the state start-up LANGU:IA (start-up d'Etat), supported by the French Ministry of Culture and DINUM, as part of the prefiguration of the service offering of the Alliance for Language Technologies EDIC (ALT-EDIC). The creation of the OpenCulture datasets was made possible by the support of LANGU:IA. The Wikidata and Wikipedia datasets were made in partnership with Wikimedia Enterprise and Wikidata/Wikimedia Germany. The corpus was stored and processed with the generous support of Genci (Jean-Zay, Idris, Eviden), Scaleway, and Tracto AI. The collection of the corpus has been largely facilitated thanks to major organizations committed to an open science approach for AI, namely AI Alliance, Mozilla, HuggingFace, Occiglot, and Eleuther AI.

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

## A  LLM USAGE STATEMENT

In the process of developing this work, we utilized LLMs for grammar correction and occasionally as a rewriting tool. In addition, we involved LLMs in the process of data visualization.

## B  CORPUS COMPOSITION

The documents in Common Corpus have the following structure:

- `identifier`: unique text identifier. In many cases, this also links to the original resource.
- `open_type`: one of the six leading collection groupings.
- `collection`: name of the sub-collection (Appendix D).
- `license`: details on the content licensing.
- `date`: date of creation of the resource, where known.
- `title`: title of the resource, when known, alternatively, the filename.
- `creator`: institution that published, collected, or curated the resource.
- `language`: automatically identified language.
- `word_count`: number of space-delimited words.
- `token_count`: number of tokens as calculated by PleIAs tokenizer.
- `text`: full text, without formatting.

In Table 3, we present the token, word, and document counts for the Common Corpus collections.

| Dataset | Documents | Words | Tokens |
|---|---|---|---|
| Open Government | 75,652,998 | 257,561,830,682 | 407,067,554,189 |
| Open Culture | 93,156,602 | 549,608,763,966 | 885,982,490,090 |
| Open Science | 19,220,942 | 147,305,783,453 | 281,193,563,789 |
| Open Code | 202,765,051 | 77,669,169,092 | 283,227,402,898 |
| Open Web | 96,165,348 | 33,208,509,065 | 73,217,485,489 |
| Open Semantic | 30,072,707 | 23,284,201,782 | 67,958,671,827 |
| Total | 517,033,648 | 1,088,638,258,040 | 1,998,647,168,282 |

Table 3: Dataset composition of Common Corpus. For each collection, we report the total number of documents, words (whitespace-separated), and tokens.

In Table 4, we present the top ten licences of Common Corpus documents.

| License type | Tokens |
|---|---|
| Public Domain | 1,138,508,375,958 |
| CC-By | 287,749,264,457 |
| MIT | 142,694,227,607 |
| CC-By-SA | 74,768,060,836 |
| Apache-2.0 | 68,750,977,037 |
| BSD-3-Clause | 18,483,944,333 |
| Open license | 10,432,513,767 |
| BSD-2-Clause | 5,497,145,480 |
| CC-BY-4.0 | 2,110,966,243 |
| CC0-1.0 | 1,877,206,195 |

Table 4: Token counts for the ten most common licenses in Common Corpus.

In Table 5, we present the top-50 languages in Common Corpus by token count. The token counts are presented in terms of our BPE tokenizer used to train the models described in Section 6, which was trained on a representative subsample of Common Corpus. To verify that our tokenizer serves as a strong baseline for token counts, we show its fertility in Appendix C.

| Language | Documents | Words | Tokens |
|---|---|---|---|
| English | 154,175,907 | 634,794,970,595 | 968,757,721,747 |
| French | 35,245,624 | 162,061,620,874 | 275,358,437,630 |
| German | 11,385,377 | 56,674,819,173 | 112,127,458,251 |
| Spanish | 6,530,094 | 26,215,767,271 | 46,514,142,421 |
| Latin | 2,367,110 | 16,189,444,325 | 36,031,591,540 |
| Italian | 3,804,052 | 13,207,129,356 | 24,681,637,575 |
| Polish | 2,640,613 | 5,086,555,167 | 12,146,688,669 |
| Greek | 844,122 | 3,796,018,483 | 11,376,498,056 |
| Portuguese | 1,756,922 | 5,234,373,473 | 10,262,747,943 |
| Russian | 2,762,818 | 3,222,919,854 | 9,439,453,633 |
| Dutch | 3,206,382 | 3,791,928,728 | 8,058,934,080 |
| Danish | 2,270,459 | 2,840,121,206 | 6,941,827,931 |
| Slovak | 683,174 | 2,320,831,403 | 5,148,967,838 |
| Czech | 946,534 | 1,966,829,784 | 4,798,558,092 |
| Indonesian | 1,023,361 | 1,660,129,567 | 4,381,878,823 |
| Estonian | 685,180 | 1,613,093,565 | 4,379,534,617 |
| Hungarian | 888,780 | 1,527,981,866 | 4,110,878,972 |
| Swedish | 3,250,289 | 1,782,642,556 | 4,014,927,806 |
| Finnish | 931,201 | 1,356,653,003 | 3,943,036,413 |
| Maltese | 480,491 | 1,421,372,608 | 3,646,102,921 |
| Bulgarian | 576,997 | 1,444,748,621 | 3,422,182,324 |
| Lithuanian | 539,906 | 1,192,586,834 | 3,097,400,907 |
| Romanian | 725,766 | 1,398,178,029 | 2,909,452,579 |
| Japanese | 1,409,956 | 204,698,439 | 2,738,872,745 |
| Arabic | 1,291,439 | 827,392,227 | 2,682,255,180 |
| Slovenian | 504,492 | 1,107,467,643 | 2,602,943,380 |
| Latvian | 364,503 | 966,005,785 | 2,579,350,119 |
| Ukrainian | 1,378,390 | 786,829,868 | 2,561,253,212 |
| Croatian | 443,669 | 984,577,685 | 2,400,762,140 |
| Chinese | 1,426,017 | 329,649,628 | 2,238,230,225 |
| Haitian Creole | 292,718 | 762,225,686 | 1,420,886,048 |
| Turkish | 572,633 | 380,551,359 | 1,206,732,266 |
| Cebuano | 6,123,694 | 598,016,557 | 1,080,404,897 |
| Norwegian Nynorsk | 1,026,745 | 405,911,677 | 1,036,914,970 |
| Irish | 139,815 | 371,175,864 | 958,688,647 |
| Castilian | 50,785 | 457,490,798 | 893,080,621 |
| Serbian | 698,037 | 292,399,267 | 822,519,328 |
| Hebrew | 343,406 | 227,869,199 | 763,029,488 |
| Catalan | 803,532 | 368,144,006 | 736,220,789 |
| Korean | 653,460 | 143,729,419 | 677,716,397 |
| Persian | 983,632 | 209,078,446 | 668,287,975 |
| Vietnamese | 1,296,789 | 267,190,147 | 593,203,801 |
| Norwegian | 628,798 | 252,133,513 | 561,694,623 |
| Armenian | 306,959 | 109,852,686 | 539,900,475 |
| Hindi | 210,984 | 127,830,222 | 463,832,707 |
| Yiddish | 59,158 | 122,100,684 | 463,526,449 |
| Welsh | 309,573 | 191,827,552 | 438,133,529 |
| Occitan | 288,195 | 195,380,666 | 357,267,635 |
| Georgian | 172,532 | 71,678,084 | 349,884,144 |
| Basque | 439,582 | 139,077,600 | 348,891,265 |

Table 5: Top-50 languages in Common Corpus by token count. Each language is presented with its number of documents, words, and tokens in the corpus. The rows are ordered by the token count.

## C  Tokenizer Details

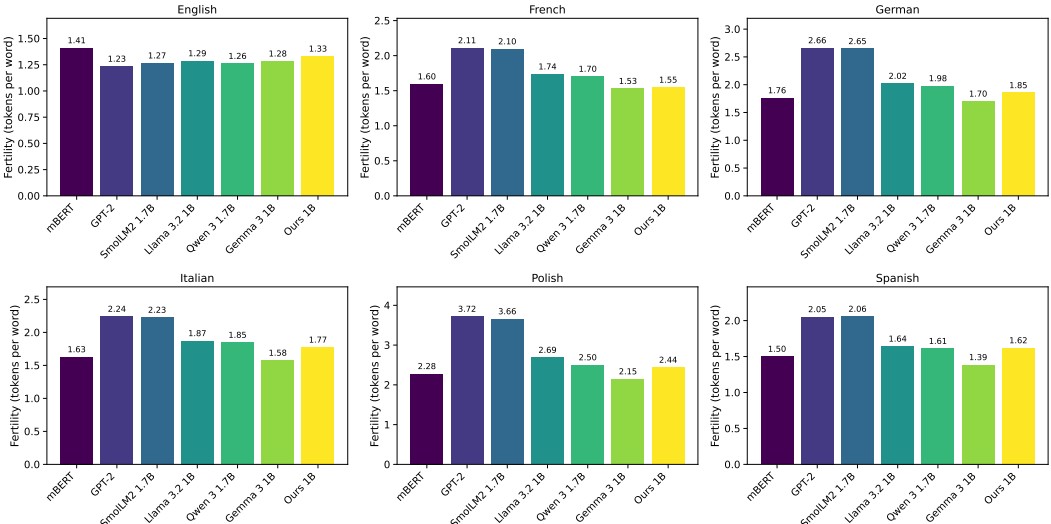

Figure 5: Comparing the fertility of PleIAs tokenizer (marked as "Ours 1B") and other language models for six languages. The data source for all languages is the `devtest` set of FLORES200 (Costa-jussà et al., 2022).

In Figure 5, we show how our tokenizer with a vocabulary of size 65,536, trained on a representative subsample of Common Corpus, compares to other language model tokenizers. Our tokenizer is outperformed only by Gemma 3, which has a tokenizer **four times larger**.

## D  Provenance

### D.1  Open Govenment

In this section, we describe the provenance and present token counts and main languages for the two sub-collections of Open Government: Finance Commons and Legal Commons.

#### D.1.1  Finance Commons

| Dataset | Main Languages | Documents | Tokens |
|---|---|---|---|
| SEC | English | 1,085,113 | 9,653,919,837 |
| WTO | English, Spanish, French, and small partitions of others | 772,508 | 2,835,007,015 |
| AMF | French, English | 595,397 | 9,823,755,281 |
| TED EU Tenders | German, French, Polish, Spanish, Dutch, Czech, Romanian, English, Swedish, Italian, Bulgarian, Finnish, Latvian, Danish, Lithuanian, Croatian, Estonian, Hungarian, Portuguese, Slovenian, Slovak, Greek, Irish | 137,837 | 650,396,761 |
| GATT Library | English, French, Spanish, Catalan, Portuguese, German | 67,596 | 224,526,628 |

Table 6: Finance Commons sources distribution with languages.

The datasets that make up Finance Commons are presented in Table 6. Here, we also present the provenance details for each of the parts of Finance Commons:

- **Securities and Exchange Commission (SEC).** This dataset comprises the SEC annual reports (Form 10-K) for the years 1993 to 2024. Entries up to 2020 were compiled by Loukas et al. (2021). We added the reports from 2021-2024, which come from the EDGAR database[6], compiled using the EDGAR-Crawler toolkit[7].

- **World Trade Organization (WTO).** This dataset comprises documents from WTO's official Documents Online platform. The documents cover the years 1995 to 2024. Documents are available in three official languages: English, French, and Spanish. Some documents are available in other languages, e.g., Chinese, Korean, Arabic, German, and Portuguese. Also released separately as WTO-PDF.

- **French Authority for Financial Market (AMF).** This is a dataset of documents from the French Authority for Financial Market, or the Autorité des marchés financiers[8] (AMF), which is an independent public authority that regulates the French market. The documents are primarily in French. Also released separately as AMF-PDF.

- **Tenders Electronic Daily (TED) EU Tenders.** This dataset is a collection of procurement notices published by the EU. The documents are published in the online version of the "Supplement to the Official Journal" of the EU[9], dedicated to European public procurement. The documents are mostly in German, with French, Polish, and Spanish making up relatively large portions of the remaining documents. There are also small portions of other languages (see details in Table 6).

- **General Agreement on Tariffs and Trade (GATT) Library.** This dataset comprises documents from GATT, which was an organization that promoted international commerce and the reduction of trade barriers among member states. Public documents were made available by the General Council of the WTO in 2006[10]. The documents span from January 1, 1946, to September 6, 1996. Most of the documents are in English, but there are also documents in French, Spanish, and other languages.

### D.1.2 LEGAL COMMONS

Here, we present the provenance details for each of the parts of Legal Commons:

- **Europarl.** This dataset is a multilingual parallel corpus, drawn from the proceedings of the European Parliament[11]. It includes texts from 21 EU languages. It was originally compiled by Koehn (2005).

- **Caselaw Access Project.** This dataset consists of 6,773,632 legal cases, digitized from Harvard Law School Library's physical collection of American case law[12]. The dataset spans the years 1658 to 2020.

- **CourtListener.** This is a dataset[13] of opinions, oral arguments, judges, judicial financial records, and federal filings put together by the Free Law Project[14].

- **EUR-lex.** This is a dataset of 57,000 legislative documents from the EU[15]. It is based on the dataset by Loza Mencía & Fürnkranz (2010) and developed by Chalkidis et al. (2019). The documents have also been annotated by the Publications Office of EU[16] with concepts from EuroVoc[17]. The dataset covers all 24 EU languages.

---

[6] https://www.sec.gov/search-filings/edgar-search-assistance/accessing-edgar-data
[7] https://github.com/nlpaueb/edgar-crawler
[8] https://www.amf-france.org/en/news-publications/publications/open-data
[9] https://ted.europa.eu/en/
[10] https://www.wto.org/english/docs_e/gattdocs_e.htm
[11] https://www.statmt.org/europarl/
[12] https://case.law/
[13] https://www.courtlistener.com/help/api/bulk-data/
[14] https://free.law/contact
[15] https://eur-lex.europa.eu/
[16] https://publications.europa.eu/en
[17] http://eurovoc.europa.eu/

| Dataset | Languages | Tokens |
|---|---|---|
| Caselaw Access Project | English | 13,821,842,995 |
| Court Listener | English | 22,625,121,735 |
| EUR-lex | Bulgarian, Croatian, Czech, Danish, Dutch, English, Estonian, Finnish, French, German, Greek, Hungarian, Irish, Italian, Latvian, Lithuanian, Maltese, Polish, Portuguese, Romanian, Slovak, Slovenian, Spanish, Swedish | 65,044,763,781 |
| Eurovoc | English, German, French, Croatian, Italian, Lithuanian, Portuguese, Finnish, Danish, Bulgarian, Dutch, Polish, Greek, Swedish, Hungarian, Czech, Spanish, Maltese, Latvian, Slovak, Slovenian, Romanian, Estonian, Arabic, Tigrinya, Farsi, Russian, Urdu, Serbian, Albanian, Kurdish, Pushto, Irish, Norwegian, Icelandic, Dari, Armenian, Japanese. | 31,648,136,898 |
| French open data | French | 24,597,392,089 |
| USPTO | English | 200,509,900,178 |
| UN Digital Library | Arabic, Chinese, English, French, Russian, Spanish | 1,781,037,875 |
| European Open Data | EU languages | 7,098,502,579 |
| OECD | English, French | 584,969,458 |

Table 7: Legal Commons sources distribution with languages.

- **Eurovoc.** Eurovoc is a dataset containing 1,528,402 documents in 39 languages with associated EuroVoc labels. The documents come from Cellar[18], which is a data repository for the Publications Office of the European Union. This dataset was originally compiled by Sébastien Campion[19].

- **French Open Data.** This dataset comes from French administrative bodies' websites, for example, the French Directorate of Legal and Administrative Information (Direction de l'information légale et administrative[20]; DILA), which is a French public administrative entity that disseminates information about laws and their applications to the public.

- **USPTO.** This dataset comprises documents from the United States Patent and Trademark Office (USPTO), the federal agency that grants patents and registers trademarks. This dataset consists of actions from this agency from 2019 to 2022. It was originally published as part of the Pile of Law (Henderson et al., 2022)[21].

- **UN Digital Library.** This dataset comes from the UN Digital Library[22].

- **European Legal Dataset.** We also collect datasets from various EU websites, *e.g.*, Archives of the EU Institute[23] and the Council of the EU[24].

---

[18] https://op.europa.eu/en/web/cellar
[19] https://huggingface.co/datasets/EuropeanParliament/Eurovoc
[20] https://echanges.dila.gouv.fr/OPENDATA/
[21] https://huggingface.co/datasets/pile-of-law/pile-of-law
[22] https://digitallibrary.un.org/?ln=en
[23] https://archives.eui.eu/
[24] https://www.consilium.europa.eu/en/general-secretariat/corporate-policies/transparency/open-data/

| Corpus | Language | Domain | Tokens |
|---|---|---|---|
| English PD | English | Books and Newspapers | 174.2B |
| US PD Books | English | Books | 82.2B |
| French PD Books | French | Books | 24.0B |
| French PD Newspapers | French | Newspapers | 110.8B |
| French PD Diverse | French | Books and Newspapers | 69.6B |
| LoC Books | English | Books | 10.6B |
| US PD Newspapers | English | Newspapers | 199.3B |
| New Zealand PD Newspapers | English, Māori | Newspapers | 12.6B |
| Europeana Newspapers | Multilingual | Newspapers | 21.0B |
| German PD Newspapers | German | Newspapers | 18.4B |
| German PD | German | Books | 58.0B |
| Portuguese PD | Portuguese | Books and Newspapers | 2.6B |
| Spanish PD Newspapers | Spanish | Newspapers | 8.0B |
| Spanish PD Books | Spanish | Books | 15.4B |
| Italian PD | Italian | Books | 18.2B |
| Dutch PD | Dutch | Books and Newspapers | 2.7B |
| BnL Newspapers | German, French, Luxembourgish | Newspapers | 0.3B |
| Danish PD | Danish | Books and Newspapers | 0.5B |
| Serbian PD | Serbian | Books and Newspapers | 0.3B |
| Czech PD | Czech | Books and Newspapers | 0.7B |
| Greek PD | Greek | Books and Newspapers | 4.2B |
| Multilingual PD | Multilingual | Books and Newspapers | 8.4B |
| Polish PD | Polish | Books and Newspapers | 5.9B |
| Latin PD | Latin | Books | 27.2B |
| Russian PD | Russian | Books | 1.9B |
| Arabic PD | Arabic | Books | 0.3B |

Table 8: Subsets of Open Culture with language coverage, type of document, and token count.

- **OECD**. These data come from the Organisation for Economic Co-operation and Development (OECD)[25].

## D.2 OPEN CULTURE

Large portion of data in Open Culture part of the Common Corpus was built on top of the following collection-as-data initiatives:

- **Chronicle America**: about 100B words (150B tokens) of digitized US newspapers by the Library of Congress, made available as a raw text file.

- **Europeana**: about 21B tokens of digitized European newspapers through large-scale cross-national contributions and new digitizations.

- **Gallica**: about 85B words of digitized French newspapers and monographs made available on the open data portal of the French digitized library through entire dumps or API access[26].

- **Biblioteca**: about 15B words of digitized Spanish newspapers and monographs.

Combined with the other retrieved data, the collections were dispatched into smaller individual subsets, which were also separately released as parts of the Open Culture collection (Table 8). The Open Culture data in Common Corpus have been post-processed and filtered, as described below, which results in a slightly different final word and token count:

---

[25]https://www.oecd.org/en/data/datasets.html?orderBy=mostRelevant&page=0
[26]https://api.bnf.fr

- **French PD.** This corpus is based on the training corpus for gallicagram[27]. It comprises 289,000 books from the French National Library (Gallica). This initial aggregation was made possible thanks to the open data program of the French National Library and the consolidation of public domain status for cultural heritage works in the EU following the 2019 Copyright Directive (Art. 14).

- **French PD Newspapers.** This dataset was also based on the Gallicagram corpus. It comprises nearly three million unique newspaper and periodical editions from the French National Library (Gallica).

- **LoC Books.** This dataset comprises 140,000 English books, digitized by the Library of Congress. The books come from the Selected Digitized Books Collection[28]. The dataset was curated by using the Library of Congress JSON API. This dataset contains only the books in the English collection. The dataset was compiled by Sebastian Majstorovic.

- **US PD Newspapers.** This dataset comprises 21 million digitized newspapers from Chronicling America[29]. The newspapers were digitized by the Library of Congress. The dataset can be fully explored through an original corpus map created by Nomic AI[30]. The dataset is mostly in English, but it also contains articles in other languages, mostly German and Spanish. The articles were published between the years 1690 and 1963.

- **New Zealand PD Newspapers.** This dataset comprises historic newspapers from New Zealand and the Pacific from the 19th and 20th centuries. The data were made available by the National Library of New Zealand as part of Papers Past[31]. The articles are primarily in English, but include some articles in te reo Māori.

- **Europeana Newspapers.** This dataset contains over 1,000 digitized newspapers from 23 libraries around Europe. It contains articles in at least 40 languages, and its articles were published between 1618 and 1990 (Neudecker, 2016). The original sources are available via Europeana, and were made available by Big Science[32].

- **German PD Newspapers.** This dataset contains articles from 4,299,653 issues from over 1900 different newspapers. The articles come from the German Digital Library, hosted by Deutsches Zeitungsportal[33]. The articles were originally published between 1794 and 1957. This dataset was curated and first made available by Sebastian Majstorovic[34].

- **German PD.** This dataset contains texts from various sources, including the Mannheim Corpus of Historical Newspapers and Magazines[35] (Mannheimer Korpus Historischer Zeitungen und Zeitschriften). This dataset is made up of 21 German newspapers and magazines. The texts were originally published between 1737 and 1905. The corpus was originally digitized between 2009 and 2011 and made available by the Institut für Deutsche Sprache in 2013.

- **Spanish PD Books.** This dataset contains 302,640 individual texts from various sources, including the leading cultural heritage institution Biblioteca Digital Hispánica[36] (BDH). To ensure that these texts are in the public domain, we have retained exclusively titles published prior to 1884.

- **Dutch PD.** This dataset contains approximately 176,000 books and 540,000 periodicals, which come from various sources including Delpher[37]. Delpher is a repository of digitized printed material from the Netherlands, which is maintained by the Koninklijke Bibliotheek, the national library of the Netherlands. To ensure that these texts are in the public domain, we have retained exclusively titles published prior to 1884.

---

[27]https://shiny.ens-paris-saclay.fr/app/gallicagram
[28]https://www.loc.gov/collections/selected-digitized-books/about-this-collection/
[29]https://chroniclingamerica.loc.gov/
[30]https://atlas.nomic.ai/data/aaron/pdnews-21286k-tr2k-addmeta/map
[31]https://paperspast.natlib.govt.nz/newspapers
[32]https://huggingface.co/datasets/biglam/europeana_newspapers
[33]https://www.deutsche-digitale-bibliothek.de/newspaper
[34]https://huggingface.co/datasets/storytracer/German-PD-Newspapers
[35]https://repos.ids-mannheim.de/fedora/objects/clarin-ids:mkhz1.00000/datastreams/CMDI/content
[36]https://www.bne.es/fr/catalogues/biblioteca-digital-hispanica
[37]https://www.digitisednewspapers.net/histories/delpher/

- **BnL Newspapers.** This dataset contains 630,709 articles from 21 different newspaper titles and 24,415 unique issues. The articles were digitized by the National Library of Luxembourg (BnL) as part of their Open Data Initiative[38]. OCR was done using Nautilus-OCR[39]. The articles are in German, French, and Luxembourgish. The newspapers were originally published between 1841 and 1879. The dataset was published and made accessible by BigScience.

- The rest of the datasets, including French PD Diverse, Portuguese PD, Italian PD, Polish PD, Danish PD, Swedish PD, Serbian PD, Czech PD, and Multilingual PD, come from various sources, including several European national libraries and cultural heritage institutions. To ensure that these texts are in the public domain, we have retained exclusively titles published prior to 1884.

### D.3 OPEN SCIENCE

In Table 9, we present the token counts inside of the Open Science collections.

| Dataset | Tokens |
|---|---|
| OpenAlex | 191,616,437,384 |
| Open Science Pile | 11,096,766,324 |
| Open Science French | 46,961,690,792 |
| Open Science Spanish | 16,523,491,767 |
| Open Science German | 7,806,446,050 |
| ArXiv | 7,188,731,472 |
| Total | 281,193,563,789 |

Table 9: Token count by dataset Open Science.

### D.4 OPEN CODE

Table 10 shows the number of tokens for the top ten coding languages and frameworks in Open Code.

| Language | Tokens |
|---|---|
| Java | 35,697,451,454 |
| JavaScript | 28,894,772,110 |
| Python | 26,681,331,771 |
| C++ | 25,481,950,314 |
| C | 23,277,000,113 |
| PHP | 23,077,121,733 |
| C# | 16,806,995,110 |
| Go | 11,200,587,099 |
| Rust | 3,888,428,173 |
| Ruby | 3,718,918,983 |

Table 10: Token counts by programming language or framework.

## E OPEN CULTURE VERIFICATION

Here, we describe the rights verification process that we applied for cultural data objects:

- **Author life + 70 years for all non-US authors.** Among most signatories of the Berne Convention for the Protection of Literary and Artistic Works[40], this is the most common

---

[38] https://data.bnl.lu/
[39] https://github.com/natliblux/nautilusocr
[40] https://www.wipo.int/treaties/en/ip/berne/

approach to determining documents in the public domain. This approach requires not only identifying the author but also their date of death. On top of the information already made available by cultural heritage institutions, we also implemented an internal data reconciliation pipeline based on the complete dump of Wikidata.

- **All publications after 1884.** In cases where the author could not be identified or for collective works like newspapers, we applied a "universal" public domain rule based on 70 years prior to the current term of the author's life + 70 years. Simplified rules like these are commonly applied in cultural heritage projects, especially for the release of newspaper collections.

- **Publication + 95 years for US authors.** This is the copyright-based approach currently in place in the US. For an international project, this will only affect US-born authors. Due to a lack of further legal expertise, we did not attempt to include works whose copyright might not have been renewed.

- **No digitization rights.** Following on the 2019 Copyright Directive (Art. 14) and common practice among GLAM reusers like Wikimedia Commons, we consider that the simple act of digitization does not provide any additional rights.

## F  CLEANING AND CURATION

We release the cleaning and curation tools as part of our *Bad Data Toolbox*, a suite designed to make challenging digitized resources usable for downstream NLP tasks and large-scale corpus construction.

### F.1  TEXT SEGMENTATION: SEGMENTEXT

**Repository:** 🤗 `PleIAs/Segmentext`

Segmentext is a token classification model trained for the structural segmentation of documents, with a specific focus on robustness to digitisation artefacts and broken or unstructured input text. In contrast to layout-based segmentation approaches that operate on visual signals such as bounding boxes or font information, Segmentext works exclusively on raw character sequences, reconstructing editorial structure from text alone. This makes it applicable in post-OCR pipelines where layout information has been lost.

The model is a DeBERTa-v2 (He et al., 2021) variant for token classification, fine-tuned on 3,500 manually annotated documents. The training data was drawn from three datasets spanning complementary domains: Open Government, Open Culture, and Open Science. This domain diversity is intended to give the model broad applicability across document types and languages.

Segmentext supports the following text segmentation:

- **Text**: the text itself.
- **Title**: any type of heading.
- **Table**: tabular content.
- **Separator**: a segmentation separator, mostly based on newline, with some variations due to text segmentation understanding.
- **Dialog**: any kind of speaker-attributed intervention.
- **Bibliography**: statement of a specific bibliographic reference, either in a bibliography section or a footnote.
- **Contact**: personal information, which can be especially useful in the context of PII removal.
- **Paratext**: any non-meaningful text included in standard documents like headers, page numbering, section recall, etc.
- **Author**: author names and signatures.
- **Date**: statement of date and time, common in letters and newspaper articles.
- **Keyword**: list of keywords, especially common in scientific publications.

Here is an example input text for the Segmentext model:

In this respect, the insurance business investment portfolio can be considered conservatively managed as it is largely composed of corporate, sovereign, and supranational bonds, term loans as well as demand deposits. Following the previous year, the group continued to diversify its holdings into investment-grade corporate bonds. It should be noted that bonds and term loans are held to maturity in accordance with the group's business model policy of "inflows".

Technical liabilities on insurance contracts.

The guarantees offered cover death, disability, redundancy, and unemployment as part of a loan protection insurance policy. These types of risk are controlled through the use of appropriate mortality tables, statistical checks on loss ratios for the population groups insured, and through the insurance program.

Liability adequacy test.

A goodness-of-fit test aimed at ensuring that insurance liabilities are adequate with respect to current statements of future cash flows generated by the insurance contracts is performed at each statement of account. Future cash flows resulting from the contracts take into account the guarantees and options inherent therein. In the event of inadequacy, the potential losses are fully recognized in the income statement. The modeling of future cash flows in the insurance liability adequacy test are based on the following assumptions: At the end of 2022, this liability adequacy test did not reveal any anomalies.

Income statement.

The income and expenses recognized for the insurance contracts issued by the group appear in the income statement in "Net income of other activities" and "Net expense of other activities".

Risk management.

The group adopts a "prudent approach" to its management of the risks to which it could be exposed through its insurance activities. Risk of counterparty. As stated above, insurance companies only invest in assets (bank deposits, sovereign bonds, supranational agencies, or corporate bonds).

Example output:

### Editorial Segmentation

[Text] In this respect, the insurance business investment portfolio can be considered conservatively managed as it is largely composed of corporate, sovereign, and supranational bonds, term loans as well as demand deposits. Following the previous year, the group continued to diversify its holdings into investment-grade corporate bonds. It should be noted that bonds and term loans are held to maturity in accordance with the group's business model policy of "inflows".

[Title] **Technical liabilities on insurance contracts.**

[Text] The guarantees offered cover death, disability, redundancy, and unemployment as part of a loan protection insurance policy. These types of risk are controlled through the use of appropriate mortality tables, statistical checks on loss ratios for the population groups insured, and through the insurance program.

[Title] **Liability adequacy test.**

[Text] A goodness-of-fit test aimed at ensuring that insurance liabilities are adequate with respect to current statements of future cash flows generated by the insurance contracts is performed at each statement of account. Future cash flows resulting from the contracts take into account the guarantees and options inherent therein. In the event of inadequacy, the potential losses are fully recognized in the income statement. The modeling of future cash flows in the insurance liability adequacy test are based on the following assumptions: At the end of 2022, this liability adequacy test did not reveal any anomalies.

[Title] **Income statement.**

[Text] The income and expenses recognized for the insurance contracts issued by the group appear in the income statement in "Net income of other activities" and "Net expense of other activities".

[Title] **Risk management.**

[Text] The group adopts a "prudent approach" to its management of the risks to which it could be exposed through its insurance activities.

[Title] **Risk of counterparty.**

[Text] As stated above, insurance companies only invest in assets (bank deposits, sovereign bonds, supranational agencies, or corporate bonds).

## F.2 OCR Error Detection: OCRoscope

**Repository:** ⬭ `PleIAs/OCRoscope`

OCRoscope is a lightweight Python library for estimating the quality of OCR-processed text without access to a ground-truth reference. Rather than relying on character- or word-level edit distance against a gold standard, OCRoscope exploits the differential behaviour of language detection models across text granularities. On long texts, language detection is robust to noise; on short character sequences, it degrades rapidly in the presence of OCR errors. The library leverages this property by applying language detection (via `pycld2`) to rolling 7-grams extracted from the input text and comparing the per-gram language assignments against the document-level language detected on the full text. The proportion of n-grams assigned to a mismatching or unknown language yields the OCR quality score: a value close to 1 indicates high-quality text, while lower values signal increasing levels of digitisation noise.

OCRoscope returns two primary metrics: a standardised OCR quality rate and a standardised rate of non-character content (punctuation, digits, whitespace anomalies). The tool supports over 80 languages via `cld2`'s language coverage. The choice of `cld2` over more accurate successors such as `cld3` is intentional: `cld2`'s greater sensitivity to short noisy sequences makes it a more discriminating signal for OCR quality estimation, yielding a score distribution that is well spread across the quality spectrum and therefore easier to threshold for downstream filtering decisions.

The tool is designed for pipeline use cases in which large corpora must be rapidly triaged: it provides a fast, reference-free quality signal that can guide decisions about whether texts warrant OCR correction, manual review, or outright exclusion.

Below we illustrate OCRoscope performance on this long text, which is correctly identified as French with >99% confidence by `cld2`. Despite the many mistakes, there are enough non-ambiguous French words:

> NOUVELLES POLI TI Q^U E S. Suede. Stockholm , le 2 5 décembre 1792. Le général Toll ira à Varsovie en quarté d'envoyé de la Suede auprès du roi et de la république ; A 1 même rey.u l'ordre de s'y rendra incessamment. 11 paraît que k Uc-régeik a des craintes ; il a fait venir chez lji les membres c Ij"' tribunal 4e la cour , et leur a rtmis son lesfca n at. La fermentation qu'a causée 1 ,'ari r?tavh n k M p v riote Thorild tî'est pas appaisée y le luigage qv'il a yailé an duc-régent a été bien entendu par le peu) k y ir M (U i n'entendrait pas l'apostrdphe suivante ? ttRxc3xa7nd >la libuk à r otre raison , et ne et nous force pas de i'ache'ef r i te n :e sang,.
>
> Le duc a fait x,épa4idre sur-le-champ une fjtbprijuun à te us les habitans di$ Toyaume , pour les detourntr de mr laisser sé luire par de fa,ux bruits et des jugemens pe rver$ , e i en même temps l'ordre a. été donné à la garnison de charger et de se tenir prête à marcher.
>
> (Mercure Français, 1793, January 25th)

Yet one short n-gram ("n k M p v riote Thorild") is classified as unknown by `cld2`. Complete processing by OCRoscope yielded a 41% rate of non-recognized 7-grams, resulting in an OCR quality rate of 59%. In comparison, the self-estimated rate of OCR-validated words by the French National Library is significantly higher (85%) for the whole document.

## F.3 OCR Error Detection: OCRerrcr

**Repository:** 🤗 `PleIAs/OCRerrcr`

OCRerrcr is a small token classification model trained to identify individual OCR errors at the token level within a text. Unlike OCRoscope, which produces a document-level quality score, OCRerrcr generates a fine-grained annotation of which tokens are likely to be erroneous, making it suitable for targeted correction workflows and for producing spatially explicit error rate estimates.

The model is a DeBERTA-v3 (He et al., 2023) with 400M parameters fine-tuned on a corpus of approximately 1,000 documents with manually labelled OCR errors, drawn from Open Government

and Open Culture. This dual-domain training is intended to give the model coverage across both modern administrative language and historical or literary language, which present different error profiles. OCRerrcr provides an accurate reference-free OCR error rate estimate, outperforming OCRoscope in precision, particularly on documents with a low or moderate error rate. However, it scales less efficiently than OCRoscope to very large corpora due to its per-token inference cost.

The model's name is itself a playful illustration of the problem it addresses: *OCRerrcr* is a plausible OCR misreading of *OCRerror*, with the vowels dropped as they might be in a degraded scan.

The following is a low-error example sentence taken from Open Culture:

> They did not approach cer, but turned away and passed irom her presence, filled with sorrow and moved with sympathy, which her intense emotions seemed to communicate to even these thoughtless young men of the tho plains.

The version with the errors highlighted by OCRerrcr is presented below:

> They did not approach <er>cer,</er> but turned away and passed <er>irom</er> her presence, filled with sorrow and moved with sympathy, which her intense emotions seemed to communicate to even these thoughtless young men of the <er>tho</er> plains.

### F.4 OCR ERROR CORRECTION: OCRONOS

**Repository:** 🤗 `PleIAs/OCRonos`

OCRonos is a generative language model trained for the correction of badly digitised text, including OCR errors, incorrect word splits and merges, and broader structural artefacts introduced during digitisation. It belongs to the corrective tier of the Bad Data Toolbox, operating downstream of quality detection tools such as OCRoscope and OCRerrcr.

The training data for OCRonos consists of a diverse multilingual set of OCR-processed texts drawn from Open Culture and Open Government. The model is fine-tuned from Llama-3-8B (Grattafiori et al., 2024).

OCRonos is designed to be conservative: it is generally faithful to the original source material and rarely rewrites words that are not erroneous. On highly degraded inputs, however, it can function as a synthetic rewriting tool rather than a strict corrector. A known issue at scale is the occasional inclusion of spuriously repeated words, which can be filtered in post-processing. The model also mitigates a common failure mode observed in smaller generalist LLMs, namely language switching: faced with heavily noisy input, models such as GPT-3.5 or Claude Haiku have been observed to drift into a different language or script during correction. OCRonos is specifically trained to resist this behaviour.

Below we present an exemplar text containing various OCR errors:

> Theguaran tees offered cover death,disability,r e dundancy andunem ployment aspartof aloanprotect ion insurance policy. These types o f risk are controlled throu ghthe use o f app ropriate morta litytables,statistica lchecksonloss rat ios for thepopulation groups insure dandthrough ar e insurance program.

The following is the version corrected by OCRonos:

> The guarantees offered cover death, disability, redundancy, and unemployment as part of a loan protection insurance policy. These types of risk are controlled through the use of appropriate mortality tables, statistical checks on loss ratios for the population groups insured, and through the insurance program.

### F.5 TOXICITY DETECTION: CELADON

**Repository:** 🤗 `PleIAs/celadon`

**Data:** 🤗 `PleIAs/ToxicCommons`

Celadon is a multilingual toxicity classifier designed specifically for the challenges posed by public-domain pre-training data: historical language, OCR noise, and multilingual content spanning nine languages. It was developed as part of the Toxic Commons project, described in Arnett et al. (2024), and was applied to filter harmful content in Common Corpus prior to its release.

The model is a fine-tune of DeBERTa-v3-small (He et al., 2023) with five independent classification heads, each corresponding to one dimension of toxicity: race and origin-based bias (including racism, xenophobia, and bias against immigration status), gender and sexuality-based bias (including sexism, homophobia, and transphobia), religious bias, ability bias (bias relating to physical, mental, or intellectual disability), and violence and abuse (graphic violence, threats, and incitement). Each head is a separate linear layer with a custom weighted cross-entropy loss function to handle class imbalance.

The model was trained on 600,000 samples drawn from ToxicCommons, a dataset of approximately 2 million annotated public-domain text segments from 13 multilingual cultural heritage collections. Annotations were generated in two stages: a small set of human annotations was used to calibrate a prompt for the instruct version of Llama 3.1-8B (Grattafiori et al., 2024), which then produced annotations at scale across five toxicity dimensions using a 0–3 severity scale. The training set was balanced to equalize non-toxic and toxic samples.

Celadon's key design rationale is efficiency: it achieves toxicity detection performance comparable to Llama 3.1-8B-Instruct while being over 40 times faster (5 minutes versus 3.4 hours per 100,000 samples), making it practical for large-scale corpus filtering. The model is optimised for the specific challenges of its target domain: historical text, OCR noise, and non-English content, and may not generalise without adaptation to web-text corpora, for which it was not designed.

# G EVALUATIONS

Figure 4 presents the distribution of six document-level quality and character composition metrics across the five open types in a 300,000-document sample of Common Corpus:

- **Character Repetition** measures the proportion of character pairs belonging to repeated character bigram sequences within a document. This value is expected to be higher for code data, due to specific syntax and varied punctuation.

- **Type-Token Ratio** is the ratio of unique word forms to the total number of tokens. It captures lexical diversity; documents with values close to 1 are typically very short, while low values can indicate highly repetitive text. For Open Government, this value is expected to be lower as regulatory texts usually contain fixed, distinct terms and repetitive formulations.

- **Top Word Proportion** is the share of all tokens accounted for by the single most frequent word in a document. This is another measure of linguistic difersity.

- **Alphanumeric Characters** denotes the fraction of characters that are letters or digits. This proportion might be lower in code due to reliance on punctuation and whitespace symbols.

- **Whitespace Characters** gives the proportion of whitespace characters, which tends to be higher in source code and tabular data than in running prose.

- **Uppercase Characters** measures the share of uppercase letters and separates Open Source content, which is densely populated with identifiers, constants, and type annotations, from Open Culture and Open Government documents that follow standard typographic conventions.

In Tables 11, 12, 13, and 14, we present per-language scores for the studied benchmarks. On MultiBLIMP, most of the scores are significantly above random (0.5); therefore, we also highlight the best and second-best scores.

| Model | Ours 350M | Gemma 3 270M | XGLM 564M | BLOOM 560M | Ours 1.2B | Gemma 3 1B | XGLM 1.7B | OLMo 1B |
|---|---|---|---|---|---|---|---|---|
| abk | 0.550 | **0.750** | 0.475 | 0.525 | 0.675 | 0.675 | 0.325 | **0.800** |
| aln | 0.733 | **0.755** | 0.709 | 0.700 | 0.728 | **0.750** | 0.675 | 0.690 |
| amh | 0.946 | 0.929 | 0.911 | **0.973** | **1.000** | 0.955 | 0.848 | 0.964 |
| apu | **0.964** | **0.964** | 0.893 | 0.786 | **0.964** | 0.929 | **0.964** | 0.893 |
| aqz | 0.214 | 0.357 | 0.429 | **0.500** | 0.429 | 0.429 | **0.714** | 0.214 |
| arb | 0.877 | 0.913 | 0.895 | **0.923** | 0.900 | **0.951** | 0.887 | 0.782 |
| azz | 0.734 | **0.744** | 0.729 | 0.720 | 0.729 | 0.758 | **0.773** | 0.686 |
| bel | **0.799** | 0.795 | 0.574 | 0.608 | 0.853 | **0.896** | 0.577 | 0.611 |
| ben | 0.571 | 0.762 | **1.000** | 0.810 | 0.762 | **0.905** | 0.857 | 0.524 |
| bho | **0.676** | 0.647 | 0.588 | 0.588 | 0.706 | **0.794** | 0.618 | 0.588 |
| bor | **0.722** | 0.631 | 0.697 | 0.610 | **0.697** | 0.627 | 0.680 | 0.668 |
| bre | **0.942** | 0.815 | 0.554 | 0.604 | 0.938 | **0.946** | 0.615 | 0.685 |
| bua | 0.680 | **0.718** | 0.670 | **0.718** | 0.670 | 0.641 | **0.699** | 0.660 |
| bul | 0.872 | 0.880 | **0.969** | 0.623 | 0.897 | 0.945 | **0.976** | 0.735 |
| cat | 0.885 | 0.852 | **0.961** | 0.950 | 0.919 | 0.931 | **0.953** | 0.735 |
| ces | **0.824** | 0.808 | 0.579 | 0.597 | 0.858 | **0.891** | 0.603 | 0.668 |
| chu | **0.670** | 0.648 | 0.582 | 0.635 | 0.659 | **0.663** | 0.593 | 0.632 |
| cym | **0.771** | 0.730 | 0.633 | 0.611 | **0.828** | 0.796 | 0.610 | 0.796 |
| dan | 0.980 | **1.000** | 0.840 | 0.800 | 0.980 | **1.000** | 0.740 | 0.940 |
| deu | **0.967** | 0.949 | 0.961 | 0.754 | 0.977 | **0.981** | 0.969 | 0.886 |
| egy | 0.409 | 0.409 | 0.409 | **0.455** | **0.500** | 0.455 | 0.409 | 0.455 |
| ell | 0.931 | 0.937 | **0.985** | 0.676 | 0.948 | 0.975 | **0.984** | 0.842 |
| eng | **0.981** | 0.979 | 0.973 | 0.960 | 0.983 | **0.987** | 0.974 | 0.984 |
| est | 0.729 | 0.699 | **0.885** | 0.561 | 0.800 | 0.800 | **0.915** | 0.587 |
| eus | 0.916 | 0.927 | **0.963** | 0.952 | 0.916 | 0.938 | **0.982** | 0.905 |
| fao | **0.707** | 0.647 | 0.509 | 0.556 | 0.772 | **0.806** | 0.552 | 0.681 |
| fas | 0.756 | **0.810** | 0.567 | 0.577 | 0.837 | **0.919** | 0.565 | 0.655 |
| fin | 0.736 | 0.744 | **0.947** | 0.562 | 0.809 | 0.893 | **0.935** | 0.645 |
| fra | **0.994** | 0.963 | 0.963 | 0.984 | **0.993** | 0.989 | 0.976 | 0.928 |
| frm | **0.997** | 0.741 | 0.745 | 0.905 | **0.997** | 0.847 | 0.820 | 0.765 |
| fro | **0.782** | 0.701 | 0.686 | 0.709 | **0.822** | 0.725 | 0.694 | 0.679 |
| gla | **0.955** | 0.924 | 0.924 | 0.939 | 0.909 | 0.924 | 0.939 | **0.970** |
| gle | 0.750 | **0.821** | 0.750 | 0.786 | 0.679 | **0.750** | **0.750** | 0.714 |
| glg | **0.849** | 0.807 | 0.798 | 0.788 | 0.879 | **0.895** | 0.789 | 0.754 |
| got | 0.630 | **0.642** | 0.588 | 0.599 | **0.645** | 0.631 | 0.580 | 0.598 |
| grc | **0.824** | 0.719 | 0.683 | 0.623 | **0.887** | 0.758 | 0.711 | 0.707 |
| guj | **1.000** | **1.000** | **1.000** | **1.000** | **1.000** | 0.857 | **1.000** | **1.000** |
| hbo | 0.711 | **0.712** | 0.683 | 0.706 | 0.737 | **0.764** | 0.629 | 0.679 |
| hbs | **0.892** | 0.848 | 0.619 | 0.603 | 0.922 | **0.929** | 0.616 | 0.723 |
| heb | **0.853** | 0.829 | 0.609 | 0.642 | **0.876** | 0.868 | 0.585 | 0.667 |
| hin | 0.854 | 0.934 | **0.975** | 0.971 | 0.916 | 0.966 | **0.977** | 0.748 |
| hit | 0.620 | **0.640** | 0.520 | 0.600 | **0.560** | 0.480 | 0.540 | 0.540 |
| hsb | **0.683** | 0.677 | 0.624 | 0.629 | 0.667 | **0.694** | 0.624 | 0.591 |
| hun | **0.928** | 0.867 | 0.728 | 0.692 | **0.938** | 0.925 | 0.686 | 0.740 |
| hye | 0.883 | **0.898** | 0.631 | 0.623 | 0.929 | **0.946** | 0.662 | 0.671 |
| hyw | **0.813** | 0.781 | 0.583 | 0.608 | **0.894** | 0.859 | 0.551 | 0.629 |
| isl | 0.710 | **0.751** | 0.653 | 0.660 | 0.767 | **0.863** | 0.636 | 0.667 |
| ita | **0.925** | 0.910 | 0.915 | 0.670 | 0.952 | **0.965** | 0.915 | 0.791 |

Table 11: Multilingual benchmarking results on MultiBLIMP (ISO 639 language codes a*–i* in alphabetical order). "Ours" refers to PleIAs models pre-trained on Common Corpus. Within each model group, the best score is in **bold**, and the second-best is underlined.

| Model | Ours 350M | Gemma 3 270M | XGLM 564M | BLOOM 560M | Ours 1.2B | Gemma 3 1B | XGLM 1.7B | OLMo 1B |
|---|---|---|---|---|---|---|---|---|
| kat | 0.931 | **0.951** | 0.917 | 0.760 | **0.951** | **0.951** | 0.809 | 0.907 |
| kaz | 0.705 | **0.792** | 0.647 | 0.682 | 0.780 | **0.844** | 0.682 | 0.688 |
| kir | **0.930** | 0.843 | 0.914 | 0.919 | 0.935 | 0.924 | 0.843 | **0.941** |
| kmr | **0.710** | 0.662 | 0.588 | 0.579 | **0.761** | 0.748 | 0.577 | 0.588 |
| koi | **0.628** | 0.488 | 0.605 | 0.558 | 0.558 | **0.651** | 0.628 | 0.605 |
| kpv | **0.641** | 0.591 | 0.553 | 0.547 | **0.700** | 0.628 | 0.581 | 0.547 |
| krl | 0.650 | 0.612 | **0.688** | 0.573 | 0.612 | 0.642 | **0.704** | 0.573 |
| kxh | **0.483** | 0.433 | 0.475 | 0.333 | 0.458 | 0.450 | 0.442 | **0.483** |
| lat | **0.874** | 0.651 | 0.578 | 0.575 | **0.925** | 0.730 | 0.568 | 0.625 |
| lav | **0.791** | 0.747 | 0.616 | 0.604 | 0.844 | **0.862** | 0.611 | 0.623 |
| lij | **0.783** | 0.744 | 0.669 | 0.638 | 0.780 | **0.807** | 0.701 | 0.665 |
| lit | **0.928** | 0.848 | 0.745 | 0.740 | **0.947** | 0.932 | 0.736 | 0.779 |
| mar | **0.737** | 0.717 | 0.735 | 0.667 | 0.713 | **0.776** | 0.726 | **0.776** |
| mdf | 0.537 | **0.622** | 0.524 | 0.537 | **0.622** | 0.585 | 0.561 | 0.500 |
| mkd | 0.923 | **0.974** | 0.769 | 0.769 | 0.821 | **1.000** | 0.590 | 0.718 |
| myv | 0.608 | **0.614** | 0.565 | 0.560 | **0.636** | 0.619 | 0.532 | 0.547 |
| nds | **0.736** | 0.729 | 0.674 | 0.663 | **0.749** | 0.732 | 0.674 | 0.700 |
| nhi | 0.526 | 0.579 | 0.474 | **0.632** | 0.553 | **0.579** | 0.447 | 0.500 |
| nld | **0.924** | 0.912 | 0.620 | 0.627 | 0.954 | **0.963** | 0.663 | 0.829 |
| olo | 0.679 | 0.668 | **0.795** | 0.611 | 0.753 | 0.711 | **0.842** | 0.595 |
| orv | **0.733** | 0.721 | 0.690 | 0.636 | **0.757** | 0.744 | 0.707 | 0.667 |
| ota | 0.879 | **0.929** | 0.899 | 0.848 | 0.939 | **0.949** | 0.889 | 0.828 |
| pcm | **1.000** | **1.000** | **1.000** | 0.923 | **1.000** | 0.962 | **1.000** | 0.885 |
| pol | **0.892** | 0.849 | 0.624 | 0.634 | 0.930 | **0.931** | 0.628 | 0.725 |
| por | 0.948 | 0.933 | 0.939 | **0.955** | 0.965 | **0.972** | 0.920 | 0.872 |
| quc | **0.779** | 0.672 | 0.649 | 0.740 | **0.740** | 0.664 | 0.679 | 0.656 |
| ron | 0.868 | **0.874** | 0.638 | 0.608 | 0.903 | **0.928** | 0.640 | 0.793 |
| rus | 0.921 | 0.916 | **0.937** | 0.727 | 0.952 | **0.963** | 0.954 | 0.819 |
| sah | 0.688 | 0.771 | 0.736 | **0.792** | 0.708 | 0.681 | 0.701 | **0.764** |
| san | 0.657 | **0.666** | 0.612 | 0.609 | 0.670 | **0.678** | 0.618 | 0.620 |
| slk | **0.797** | 0.739 | 0.528 | 0.570 | 0.824 | **0.861** | 0.533 | 0.588 |
| slv | **0.882** | 0.796 | 0.618 | 0.636 | **0.903** | 0.854 | 0.622 | 0.711 |
| sme | 0.689 | **0.705** | 0.653 | 0.660 | 0.681 | **0.700** | 0.668 | 0.659 |
| sms | **0.833** | 0.802 | 0.779 | 0.757 | **0.821** | 0.779 | 0.764 | 0.768 |
| spa | 0.959 | 0.945 | 0.950 | **0.966** | 0.970 | **0.973** | 0.956 | 0.896 |
| sqi | 0.786 | **0.823** | 0.494 | 0.588 | 0.823 | **0.881** | 0.539 | 0.765 |
| swe | **0.995** | **0.995** | 0.970 | 0.950 | 0.995 | **1.000** | 0.985 | 0.990 |
| tam | 0.942 | 0.942 | **0.969** | 0.966 | 0.932 | 0.953 | **0.976** | 0.746 |
| tpn | **0.111** | 0.000 | **0.111** | **0.111** | **0.222** | 0.111 | 0.000 | 0.000 |
| ttc | 0.478 | 0.478 | **0.493** | 0.449 | 0.449 | 0.464 | 0.435 | **0.478** |
| tur | 0.766 | 0.804 | 0.829 | 0.700 | 0.814 | **0.908** | 0.857 | 0.716 |
| uig | **0.764** | 0.760 | 0.722 | 0.732 | 0.755 | **0.757** | 0.686 | 0.740 |
| ukr | 0.874 | **0.892** | 0.640 | 0.606 | 0.911 | **0.946** | 0.648 | 0.704 |
| urb | **0.538** | 0.462 | 0.462 | 0.231 | **0.538** | 0.462 | 0.462 | 0.462 |
| urd | 0.858 | 0.925 | **0.956** | 0.935 | 0.896 | **0.964** | 0.960 | 0.736 |
| uzb | **0.900** | 0.880 | 0.780 | 0.720 | 0.900 | **0.940** | 0.760 | 0.880 |
| vep | 0.572 | **0.588** | **0.588** | 0.481 | **0.631** | 0.626 | 0.567 | 0.524 |
| wbp | **0.250** | 0.000 | 0.167 | **0.250** | **0.250** | 0.083 | **0.250** | **0.250** |
| wol | **0.892** | 0.881 | 0.851 | 0.891 | 0.868 | **0.879** | 0.854 | 0.823 |
| xcl | **0.679** | 0.674 | 0.585 | 0.636 | **0.711** | 0.702 | 0.613 | 0.616 |
| xnr | **0.791** | 0.756 | 0.779 | 0.616 | 0.744 | **0.814** | 0.721 | 0.733 |
| xpg | 0.820 | **0.900** | 0.820 | 0.860 | 0.880 | **0.900** | **0.900** | **0.900** |
| yrl | 0.689 | **0.722** | 0.594 | 0.604 | **0.683** | 0.669 | 0.626 | 0.601 |

Table 12: Multilingual benchmarking results on MultiBLIMP (ISO 639 language codes k*–y* in alphabetical order). "Ours" refers to PleIAs models pre-trained on Common Corpus. Within each model group, the best score is in **bold**, and the second-best is underlined.

| Model | Ours 350M | Gemma 3 270M | XGLM 564M | BLOOM 560M | Ours 1.2B | Gemma 3 1B | XGLM 1.7B | OLMo 1B |
|---|---|---|---|---|---|---|---|---|
| ar | 0.475 | 0.492 | 0.500 | 0.521 | 0.477 | 0.572 | 0.525 | 0.473 |
| ca | 0.514 | 0.513 | 0.567 | 0.561 | 0.535 | 0.600 | 0.602 | 0.509 |
| en | 0.569 | 0.614 | 0.606 | 0.612 | 0.617 | 0.698 | 0.645 | 0.704 |
| es | 0.520 | 0.558 | 0.549 | 0.555 | 0.543 | 0.628 | 0.593 | 0.556 |
| eu | 0.516 | 0.524 | 0.531 | 0.538 | 0.514 | 0.531 | 0.561 | 0.503 |
| gl | 0.490 | 0.486 | 0.461 | 0.467 | 0.532 | 0.576 | 0.484 | 0.459 |
| hi | 0.509 | 0.541 | 0.520 | 0.549 | 0.515 | 0.598 | 0.557 | 0.493 |
| id | 0.500 | 0.544 | 0.542 | 0.555 | 0.518 | 0.631 | 0.583 | 0.498 |
| my | 0.492 | 0.507 | 0.515 | 0.475 | 0.498 | 0.518 | 0.537 | 0.475 |
| ru | 0.503 | 0.547 | 0.562 | 0.488 | 0.531 | 0.634 | 0.600 | 0.512 |
| sw | 0.500 | 0.507 | 0.531 | 0.500 | 0.510 | 0.551 | 0.563 | 0.494 |
| te | 0.542 | 0.562 | 0.559 | 0.557 | 0.553 | 0.592 | 0.581 | 0.532 |
| zh | 0.491 | 0.536 | 0.533 | 0.545 | 0.494 | 0.594 | 0.561 | 0.511 |

Table 13: Multilingual benchmarking results on XStoryCloze. "Ours" refers to PleIAs models pre-trained on Common Corpus. Languages are represented as two-letter codes in ISO 639.

| Model | Ours 350M | Gemma 3 270M | XGLM 564M | BLOOM 560M | Ours 1.2B | Gemma 3 1B | XGLM 1.7B | OLMo 1B |
|---|---|---|---|---|---|---|---|---|
| es | 0.566 | 0.590 | 0.604 | 0.620 | 0.614 | 0.678 | 0.664 | 0.524 |
| et | 0.518 | 0.498 | 0.554 | 0.488 | 0.500 | 0.536 | 0.568 | 0.480 |
| eu | 0.504 | 0.514 | 0.512 | 0.502 | 0.518 | 0.502 | 0.534 | 0.516 |
| ht | 0.522 | 0.504 | 0.548 | 0.500 | 0.524 | 0.518 | 0.556 | 0.534 |
| id | 0.534 | 0.578 | 0.574 | 0.596 | 0.558 | 0.690 | 0.646 | 0.544 |
| it | 0.542 | 0.524 | 0.536 | 0.502 | 0.562 | 0.648 | 0.536 | 0.488 |
| qu | 0.522 | 0.492 | 0.492 | 0.500 | 0.500 | 0.502 | 0.522 | 0.506 |
| sw | 0.528 | 0.546 | 0.530 | 0.516 | 0.538 | 0.544 | 0.562 | 0.510 |
| ta | 0.536 | 0.560 | 0.562 | 0.558 | 0.556 | 0.566 | 0.550 | 0.550 |
| th | 0.546 | 0.538 | 0.550 | 0.538 | 0.550 | 0.584 | 0.580 | 0.532 |
| tr | 0.530 | 0.566 | 0.544 | 0.528 | 0.542 | 0.606 | 0.536 | 0.530 |
| vi | 0.550 | 0.598 | 0.584 | 0.602 | 0.526 | 0.694 | 0.630 | 0.494 |
| zh | 0.536 | 0.564 | 0.554 | 0.588 | 0.544 | 0.640 | 0.584 | 0.522 |

Table 14: Multilingual benchmarking results on XCopa. "Ours" refers to PleIAs models pre-trained on Common Corpus. Languages are represented as two-letter codes in ISO 639.

