# OpenReview forum: "Common Corpus: The Largest Collection of Ethical Data for LLM Pre-Training"
_ICLR.cc/2026/Conference — ICLR 2026 Oral_

### Official Review · Reviewer_GQ3w · 2025-10-29

**Soundness:** 3
**Presentation:** 3
**Contribution:** 3
**Rating:** 6
**Confidence:** 4

**Summary:**

The paper introduces Common Corpus, a 2T-token multilingual dataset built entirely from open or permissively licensed sources (public domain, CC, open code). It aims to offer a legally clean, transparent alternative to web-scraped corpora. The authors describe the collection process, cleaning (PII removal, OCR correction, toxicity filtering), and train two small models showing comparable multilingual performance to baselines.

**Strengths:**

- Timely and relevant: strong contribution to open and compliant LLM research.
- Impressive scale (2T tokens) and careful documentation of provenance and licenses.
- Multilingual coverage beyond English, rare for open corpora.
- Clear adherence to emerging best-practice frameworks (e.g., dataset documentation, PII filtering).
- Demonstrates feasibility through working models and released tools.

**Weaknesses:**

- Empirical section is limited, only small models and a few benchmarks.
- No clear comparison to similarly “open” corpora (e.g., Dolma, KL3M) in terms of quality or coverage.
- Curation process, though detailed, lacks quantitative measures of data quality after filtering.
- Language balance is heavily skewed to English (~50%).
- Evaluation of ethical filtering (toxicity, PII accuracy) could be better substantiated.

**Questions:**

- How scalable is the current pipeline to truly support trillion-token multilingual expansion?
- How do the authors ensure consistent quality across OCRed historical data?
- Could releasing the filtering tools lead to reproducibility or bias-transfer risks?
- Do they plan to release validation splits or subsets for standardized benchmarking?

---

> ### Author Response · Authors · 2025-11-18
> **Response to the Review**
>
> We thank the reviewer for finding our work relevant and impressive and for giving valuable feedback.
>
> In Section 5, we show the models of sizes comparable to the models we trained ourselves. We chose SOTA closed-data models, such as Gemma 3, as well as other notable open-data models, such as BLOOM and OLMo. The benchmarks were chosen according to two criteria: (a) they do not require specific instruction tuning and can be fairly applicable to base models, and (b) they highlight multilingual performance without machine-translated or generated data. This also implicitly compares our corpus to the Dolma and ROOTS datasets. Additionally, we provide a broader comparison to other open corpora in the **General Comment** above. We will include the details in the Related Work section in a revised paper.
>
> > Language balance is heavily skewed to English (~50%).
>
> The highest source language in the Common Corpus is English due to the availability of the open data. However, as the reviewer mentions in the **Strengths** section, it is rare for open corpora to have large multilingual support (see the comparison in the **General Comment** above).
>
> > How scalable is the current pipeline to truly support trillion-token multilingual expansion?
>
> Given that the current pipeline has already been applied to 2 trillion tokens, it is scalable for such orders of magnitude. The main scaling constraint now is more about new data collection rather than processing. Some parts of our open pipelines have been successfully applied to other large-scale projects, which we cannot link in this rebuttal to preserve anonymity.
>
> > How do the authors ensure consistent quality across OCRed historical data?
>
>
> We developed a new referenced framework for low OCR quality filtering, OCRoscope, which has been adopted by other prominent dataset curators (we will not add a reference to preserve anonymity). OCRoscope relies on comparing the language detection of large segments with the language detection of short ngrams, which is much more affected by OCR noise. We removed the lowest-quality samples, which mostly come from Newspapers. This ensures the removal of at least the lowest-quality data.
>
> We will add plots comparing empirical evaluations of our corpus data compared to other open corpora, using such metrics as repetition counts. However, common quality measures such as perplexity are model-specific and would be unsuitable for Common Corpus due to its unique inclusion of historical data, which would likely be spuriously labeled as low-quality by existing models. The same applies to low-resource languages. Furthermore, across languages, data quality tools vary in quality, reflecting known data quality issues [1].
>
> > Could releasing the filtering tools lead to reproducibility or bias-transfer risks?
>
> The release of processing tools largely improves reproducibility because these are the same tools we applied to the raw data. As for the bias-transfer risks, it is possible that our filtering methods introduce bias. By describing our methods and releasing our tools, we make it possible to audit in the future and evaluate potential bias. We leave this to future work.
>
> > Do they plan to release validation splits or subsets for standardized benchmarking?
>
> As we position Common Corpus as a pre-training corpus, we are not planning to separately provide validation or benchmarking splits. To the best of our knowledge, it is not a common practice among similar initiatives, such as Dolma and ROOTS.
>
> [1] Julia Kreutzer et al. “Quality at a Glance: An Audit of Web-Crawled Multilingual Datasets”, 2022

---

### Official Review · Reviewer_LPfF · 2025-10-30

**Soundness:** 3
**Presentation:** 3
**Contribution:** 3
**Rating:** 6
**Confidence:** 4

**Summary:**

This paper introduces the Common Corpus, an open dataset for pre-training LLMs. It is constituted of texts that are either uncopyrighted or under permissible licenses, for a total of two trillion tokens across a variety of languages and tasks such as coding, as well as a diversity in terms of regions and time.

The authors go into the data collection and curation process in great detail, and also train two LLMs of small size on the dataset that show that they perform on a similar level to other models of their size across a variety of datasets.

**Strengths:**

Developing an explicitly curated dataset based on data licensing is important and a crucial contribution to the AI community.

There is an emphasis on diversity in terms of languages and regions.

The code used for creating the dataset is available, allowing others to reproduce it.

The resulting dataset can be filtered based on different criteria, including license and language, which makes it useful for developers and researchers working on specific languages or historical periods.

Personally Identifiable Information is removed with the Presidio tool, which means that there is almost no risk of data leakage from the trained models.

**Weaknesses:**

- "Ethical data" is a very relative/hard-to-define concept -- maybe "consensual data" or "legal data" would be better alternatives?

- Figure 1 is hard to read because languages such as English, Spanish or French are spoken in multiple places, so simply putting a dot on Madrid or Paris isn't representative of where the language is from

- "synthetically rewrite the document without the harmful language" - how is this done and verified? Doesn't introducing synthetically-generated text dilute the corpus?

**Questions:**

- How are the six collections defined, what are the criteria?

- How are you sure (are you sure) that none of the data is LLM-generated?

- Are the audio transcripts AI- or human- generated?

- How has the set of Wikidata been adapted in natural language? The example provided isn't very clear

- "Segmentext should work correctly on diverse document formats" - did you do testing? In general, providing more information about the tools that you developed and how they work would help understand their limitations and applicabliity.

**Details Of Ethics Concerns:**

I am not a lawyer, so it would be good to have a second opinion about the types of licenses that have been kept for the corpus, and whether they are indeed permissible for training LLMs on.

---

> ### Author Response · Authors · 2025-11-18
> **Response to the Review**
>
> We are grateful to the reviewer for valuable feedback and for highlighting the strengths of our work.
>
> Though the word “ethical” might sound underdefined, after careful consideration, we believe that this is the most accurate term. It is already used in other data initiatives to encompass the practices in data acquisition and curation. “Legal” might incorrectly imply that the data is mostly about court cases (as in our Open Government collection), and “consensual” is also not necessarily accurate, as we do not have explicit author consent. In all, the word “ethical” additionally implies a broader multilingual coverage, expansion of data sources beyond the web, and careful treatment of PII and toxic content.
>
> Figure 1 is primarily intended to demonstrate a general overview rather than to provide detailed statistics. As we also mention in our reply to Reviewer aTGr, we show more details in the **General Comment** above and will include a full language distribution in a separate Appendix chapter in the updated manuscript.
>
> > Doesn't introducing synthetically-generated text dilute the corpus?
>
> We believe that the impact of synthetic rewriting is minor, especially relative to the gains in toxicity mitigation, and given that the ideas behind the original data points remain unique. Approximately 5% of the public domain data was synthetically rewritten.
>
> > How are the six collections defined, what are the criteria?
>
> The collections are defined by their primary domains and the specifics of data sources, which we describe in corresponding subsections 3.1-3.6. This allows for filtering in broad-domain terms when, e.g., only the data from public institutions is needed (hence Open Government). Should one need to select a more specific partition, there are more granular subcollections based on the data source, as described in the Appendix tables, as well as the language tags.
>
> > How are you sure (are you sure) that none of the data is LLM-generated? Are the audio transcripts AI- or human-generated?
>
> Audio transcripts are generated with Whisper. We note that an independent validation of our curation approach has been the training of text-to-speech models, which currently lead the HuggingFace leaderboard for speech recognition. We will not link these to preserve anonymity.
>
> Regarding the rest of the corpus, detailed provenance ensures the attribution and reliability of the data. The large parts of the historical data come from OCR-processed historical documents, which are unlikely to be generated; the same is expected for the governmental and law data that come from official sources. The parts that might contain generated content are scientific articles, as we might expect a certain partition of arXiv papers to be LLM-generated to a certain extent, as well as the open-source GitHub code. However, our current cut-off date is mid-2024; therefore, we expect the rate of generated content to be low.
>
> > How has the set of Wikidata been adapted in natural language? The example provided isn't very Clear
>
> For Wikidata, we mapped the identifier and properties to their corresponding natural labels in all available languages. The data is presented in a list-like format and does not incorporate any synthetic rewriting. This integration was primarily motivated by multilingual alignment, as many facts and information do not have corresponding articles in local versions of Wikipedia.
>
> > "Segmentext should work correctly on diverse document formats" - did you do testing?
>
> While developing the model, we ran through several iterations with manual testing and validation on different input data formats, but did not conduct a formal evaluation. Due to being an encoder and not requiring the original PDF sources, SegmenText was a cost-effective solution at scale for pretraining, at least when re-digitization using a modern VLM is not affordable. We will document the developed tools better in the revised version of the paper.

---

> > ### Comment · Reviewer_LPfF · 2025-11-25
> >
> > Thank you to the authors for their response.
> >
> > I still believe that 'ethical' (in the context of a scientific paper) is not necessarily the right choice of terminology, but I understand why it was made.

---

### Official Review · Reviewer_BY7t · 2025-11-01

**Soundness:** 3
**Presentation:** 3
**Contribution:** 4
**Rating:** 8
**Confidence:** 3

**Summary:**

This paper accompanies the release of the largest open dataset for LLM pretraining: the Common Corpus. The authors fully emphasize the “open” aspect of their data, providing full transparency around the provenenance, source, and licensing of their data. They also engage in extensive data cleaning and quality improvement steps for different segments of their data. Unlike its recent open data precusors, Common Corpus also includes substantial multilingual data.

**Strengths:**

The writing of this paper was very clear, and this work was done with much care, detail, and rigor. For example, the authors “involve local communities” in gathering data from diverse sources and did not machine-translate the multilingual component of their dataset. This resource is an invaluable contribution and far exceeds its predecessors in size and composition. Aside from data, this work also contributes several data cleaning tools.

**Weaknesses:**

I understand that the main focus of this paper is on the data and not on model training, so it is okay that the model training results aren’t groundbreaking. One minor detail is that your choice of benchmarking results to show in Section 5 is a little strange. You focus on a few multilingual benchmarks, but compare against OLMo 1B, which may have been intended to be monolingually English.

It would have been nice to see a clear tabulation (like, a table) of how this dataset overlaps, extends, or differs from existing open datasets.

**Questions:**

Data may be copied and then mislicensed. Have you checked for overlap between your dataset and data that has less permissive licenses? This might be really tricky to do, so I am just curious.

Is there a way for someone to remove data about themselves or produced by themselves from this dataset? Right to be forgotten, etc.

---

> ### Author Response · Authors · 2025-11-18
> **Response to the Review**
>
> We thank the reviewer for a thorough review and highly appreciate their interest in our work.
>
> We present the results of model training to demonstrate that training on fully open data can lead to comparable or, as in the case of MultiBLIMP, better performance than for the closed-data training. Among other models, we also compare to OLMo 1B as an example of another model trained on a fully open and published corpus (Dolma). Although OLMo was not intended to be highly multilingual, its score on MultiBLIMP is well above random, which suggests that the model possesses multilingual capabilities.
>
> Regarding the overlaps with other open corpora, for the purpose of this rebuttal, we focused our comparison on Fineweb, given its centrality in open LLM research. Evaluations of the top 1000 domains show a very low overlap with Common Corpus: less than 2% of pages and 1% of domain names have some content in common. This includes Wikipedia, Wikisource, StackOverflow, ChronicleAmerica (with uncorrected OCR on fineweb), SEC, Caselaw, and several scientific websites (Springer, Pubmed, Wiley, etc.). Even within the overlapping collections, content can differ significantly, as we focused on large PDF sets while agnostic crawl pipelines will mostly retrieve HTML pages (a typical example being abstracts from Scientific papers). This means that Common Corpus mostly adds new, diverse content to the open pretraining ecosystem and significantly differs from crawl corpora, which we will further stress in the final version of the paper.
>
> > Data may be copied and then mislicensed. Have you checked for overlap between your dataset and data that has less permissive licenses? This might be really tricky to do, so I am just curious.
>
> We limited our comparative analysis between Common Corpus and crawled datasets at a metadata level: text reuse detection is highly challenging given the prevalence of copyright relicensing of open content by third parties (e.g., copyfraud). We believe that our curation-based approach, focused on content made available by large institutions/infrastructures, largely addresses this concern.
>
> > Is there a way for someone to remove data about themselves or produced by themselves from this dataset?
>
> We are responsive to requests to remove data from our corpus. For example, we already removed some Dutch Public Library data at their request due to uncertainty over the public domain status, e.g., due to uncertainty of the author's date of death or specific status of collective work (newspapers) in Dutch Law. For other subsets of the data, because the datasets are mostly curated from large institutional data, we will be receptive to requests from those organizations.

---

### Official Review · Reviewer_aTGr · 2025-11-03

**Soundness:** 3
**Presentation:** 2
**Contribution:** 3
**Rating:** 8
**Confidence:** 5

**Summary:**

The paper introduces Common Corpus, a ~2T-token multilingual dataset  for LLM pre-training that contains only data that are in the public domain or released under permissive open licenses.  The authors document data sources, licensing, cleaning tools (Segmentext, OCRonos, Celadon, PII filtering), and present benchmark results for small Llama-style models trained solely on this corpus.

Overall, this is a valuable contribution to open LLM research. The motivation is strong, and the authors make extensive efforts to demonstrate the usefulness of the dataset for pre-training purposes.

However, several aspects including the very Euro-centric nature of the multilingual coverage and lack of quantitative information about ambiguous licenses need further clarification.

**Strengths:**

1. Timely contribution to open-data infrastructure for LLMs, especially given that many existing datasets had to be taken down due to license issues.

2. Transparent and detailed documentation of dataset provenance, content, and filtering pipeline

3. Releases and discusses useful tools for OCR correction and sentence segmentation

4. Demonstrates that open data sourced from only permissive licenses can still yield competitive model performance.

**Weaknesses:**

1. The claim of “multilingual diversity” is overstated as the dataset is heavily Euro-centric. The top ten languages are all European, and there is little inclusion of African or Asian languages.

2. It is not clear how trustworthy the licensing information is. The authors do not report how many documents had ambiguous, conflicting, or missing license information.

3. For certain types of data, licenses can apply inconsistently across sub-components. It is not clear how these were disambiguated.

4. The data pre-processing pipeline, including Segmentext and OCRonos are underdescribed. It is not clear if any de-duplication was done or required.

5. The evaluation lacks detail on per-language performance and broader benchmark coverage.

**Questions:**

1. Can the authors provide a full language inventory and token count distribution, including low-resource languages?

2. How are ambiguous or missing licenses handled, and what proportion of the dataset do they represent?

3. What steps were taken to ensure deduplication across overlapping sources?

---

> ### Author Response · Authors · 2025-11-18
> **Response to the Review**
>
> We thank the reviewer for highlighting the strengths and timeliness of our work.
>
> **Multilingual Diversity.** We do agree that having more non-European data in our corpus would strengthen our claim about multilingual diversity. However, Common Corpus still presents much more linguistic diversity compared to other available open corpora, such as Dolma or Common Pile, which are English-centric. We present a comparison table in the **General Comment** above and will also include these figures in the updated manuscript. One benefit of focusing on European languages is that we are most familiar with European copyright laws; therefore, we are best able to determine what constitutes permissive licenses in this jurisdiction. In the future, we hope to expand our dataset curation efforts beyond European languages.
>
> **License Handling.** We generally constrained the dataset collection to cases where licenses could be determined in bulk, thereby mostly excluding web data in general. For subsets like YouTube, we relied on the licenses provided by content owners. The main source of uncertainty was the public domain, as copyright expiration varies significantly across countries. We exclusively included content classified as public domain by major cultural heritage institutions, but applied additional filters to further limit risks based on author death dates and publication dates, and removed content with insufficient metadata.
>
> **Deduplication.** Most of Common Corpus is taken from extant works in a PDF format. Our early experiments showed a negligible rate of duplication. This has been independently confirmed by follow-up work, which we will not link to preserve anonymity. We attribute this low rate of duplication to our curation-based approach: large institutions are already incentivized to avoid re-digitizing the same texts. We still raise a concern over the resiliency of existing deduplication tools to the variations introduced by digitization artifacts.
>
> The code data has a higher risk of duplicates, so to ensure the deduplication, we took the deduplicated version of the Stack to form this part of the corpus. We will include additional details in the processing section of the revised paper.
>
> **Model Benchmarking.** Small models struggle with many task formats, due to the need to both solve the task and correctly handle the format of the question ("auxiliary task demands" [1]). We selected tasks that could be evaluated using a log-likelihood/cloze style. We also evaluated the models on Belebele, a QA benchmark available for many languages. However, we found that none of the models performed significantly above chance. Therefore, we did not include these results. In response to the reviewer’s request, we also include a per-language breakdown in the **General Comment** above.
>
> [1] Jennifer Hu and Michael Frank “Auxiliary task demands mask the capabilities of smaller language models”

---

### Author Response · Authors · 2025-11-18
**General Comment [1/2]**

We are grateful to all reviewers for their constructive feedback and valuable suggestions. In this general comment, we would like to present several tables to reference in our replies.

First, we show the general comparison of Common Corpus to other popular open corpora across the dimensions of licensing, multilinguality, domain diversity (e.g., whether the corpus is only code or only governmental data vs corpora combining multiple diverse domains), and reliance on data crawled from the web. As the table shows, Common Corpus is the only open dataset combining all factors.


| Dataset  | DCAD-2000 | KL3M | ROOTS | Common Pile | FineWeb | C4 | Dolma | Common Corpus |
|------------------|-----------|------|-------|-------------|---------|----|-------|---------------|
| Permissive data     |    N       | Y    | N     | Y           |    N   |   N    | N       | Y      |
| Multilingual             | Y         | N    | Y     | N           |     Y    |    N    | N     | Y        |
| Multidomain            | Y         | N    |   Y     | Y           |   Y      |     Y   |  Y       | Y        |
| Beyond Web Crawl | N        |  Y    |  N     | Y           | N      | N     |    Y     | Y        |


In addition, to provide more details on multilinguality, we present the distribution of languages in the top-30 (top-10 are shown in Table 2 of the paper):

| Language (11-20) | # Tokens | Language (21-30) | #Tokens |
|------------------|----------|------------------|---------|
| Dutch            | 8.1B     | Bulgarian        | 3.4B    |
| Danish           | 6.9B     | Lithuanian       | 3.1B    |
| Slovak           | 5.1B     | Romanian         | 2.9B    |
| Czech            | 4.8B     | Japanese         | 2.7B    |
| Indonesian       | 4.4B     | Arabic           | 2.7B    |
| Estonian         | 4.4B     | Slovenian        | 2.6B    |
| Hungarian        | 4.1B     | Latvian          | 2.6B    |
| Swedish          | 4.0B     | Ukrainian        | 2.6B    |
| Finnish          | 3.9B     | Croatian         | 2.4B    |
| Maltese          | 3.6B     | Chinese          | 2.2B    |

---

> ### Author Response · Authors · 2025-11-18
> **General Comment [2/2]**
>
> Finally, to highlight the impact of Common Corpus on the multilingual capabilities of the model, we present the per-language scores on MultiBLIMP for a subset of languages. We will report the scores for the remaining languages in the updated manuscript:
>
> |  Model     | Ours   | Gemma 3   | XGLM   | BLOOM   | Ours   | Gemma 3   | XGLM   | OLMo   |
> |:---------------|:-------|:----------|:-------|:--------|:-------|:----------|:-------|:-------|
> | [Parameters]    | [350M]   | [270M]      | [564M]   | [560M]   | [1.2B]   | [1B]        | [1.7B]   | [1B]     |
> | Albanian | 0.733  | 0.755     | 0.709  | 0.7     | 0.728  | 0.75      | 0.675  | 0.69   |
> | Arabic | 0.877  | 0.913     | 0.895  | 0.923   | 0.9    | 0.951     | 0.887  | 0.782  |
> | Basque | 0.916  | 0.927     | 0.963  | 0.952   | 0.916  | 0.938     | 0.982  | 0.905  |
> | Belarusian | 0.799  | 0.795     | 0.574  | 0.608   | 0.853  | 0.896     | 0.577  | 0.611  |
> | Breton | 0.942  | 0.815     | 0.554  | 0.604   | 0.938  | 0.946     | 0.615  | 0.685  |
> | Bulgarian | 0.872  | 0.88      | 0.969  | 0.623   | 0.897  | 0.945     | 0.976  | 0.735  |
> | Catalan | 0.885  | 0.852     | 0.961  | 0.95    | 0.919  | 0.931     | 0.953  | 0.735  |
> | Czech | 0.824  | 0.808     | 0.579  | 0.597   | 0.858  | 0.891     | 0.603  | 0.668  |
> | Church Slavic | 0.67   | 0.648     | 0.582  | 0.635   | 0.659  | 0.663     | 0.593  | 0.632  |
> | Danish | 0.98   | 1.0       | 0.84   | 0.8     | 0.98   | 1.0       | 0.74   | 0.94   |
> | Dutch | 0.924  | 0.912     | 0.62   | 0.627   | 0.954  | 0.963     | 0.663  | 0.829  |
> | German | 0.967  | 0.949     | 0.961  | 0.754   | 0.977  | 0.981     | 0.969  | 0.886  |
> | Greek | 0.931  | 0.937     | 0.985  | 0.676   | 0.948  | 0.975     | 0.984  | 0.842  |
> | English | 0.981  | 0.979     | 0.973  | 0.96    | 0.983  | 0.987     | 0.974  | 0.984  |
> | Estonian | 0.729  | 0.699     | 0.885  | 0.561   | 0.8    | 0.8       | 0.915  | 0.587  |
> | Faroese | 0.707  | 0.647     | 0.509  | 0.556   | 0.772  | 0.806     | 0.552  | 0.681  |
> | Finnish | 0.736  | 0.744     | 0.947  | 0.562   | 0.809  | 0.893     | 0.935  | 0.645  |
> | French | 0.994  | 0.963     | 0.963  | 0.984   | 0.993  | 0.989     | 0.976  | 0.928  |
> | Old French | 0.782  | 0.701     | 0.686  | 0.709   | 0.822  | 0.725     | 0.694  | 0.679  |
> | Old Greek | 0.824  | 0.719     | 0.683  | 0.623   | 0.887  | 0.758     | 0.711  | 0.707  |
> | Hebrew | 0.853  | 0.829     | 0.609  | 0.642   | 0.876  | 0.868     | 0.585  | 0.667  |
> | Hungarian | 0.928  | 0.867     | 0.728  | 0.692   | 0.938  | 0.925     | 0.686  | 0.74   |
> | Icelandic | 0.71   | 0.751     | 0.653  | 0.66    | 0.767  | 0.863     | 0.636  | 0.667  |
> | Italian | 0.925  | 0.91      | 0.915  | 0.67    | 0.952  | 0.965     | 0.915  | 0.791  |
> | Latin | 0.874  | 0.651     | 0.578  | 0.575   | 0.925  | 0.73      | 0.568  | 0.625  |
> | Latvian | 0.791  | 0.747     | 0.616  | 0.604   | 0.844  | 0.862     | 0.611  | 0.623  |
> | Lithuanian | 0.928  | 0.848     | 0.745  | 0.74    | 0.947  | 0.932     | 0.736  | 0.779  |
> | Persian | 0.756  | 0.81      | 0.567  | 0.577   | 0.837  | 0.919     | 0.565  | 0.655  |
> | Polish | 0.892  | 0.849     | 0.624  | 0.634   | 0.93   | 0.931     | 0.628  | 0.725  |
> | Portuguese | 0.948  | 0.933     | 0.939  | 0.955   | 0.965  | 0.972     | 0.92   | 0.872  |
> | Russian | 0.921  | 0.916     | 0.937  | 0.727   | 0.952  | 0.963     | 0.954  | 0.819  |
> | Spanish | 0.959  | 0.945     | 0.95   | 0.966   | 0.97   | 0.973     | 0.956  | 0.896  |
> | Swedish | 0.995  | 0.995     | 0.97   | 0.95    | 0.995  | 1.0       | 0.985  | 0.99   |
> | Ukrainian | 0.874  | 0.892     | 0.64   | 0.606   | 0.911  | 0.946     | 0.648  | 0.704  |
> | Uzbek | 0.9    | 0.88      | 0.78   | 0.72    | 0.9    | 0.94      | 0.76   | 0.88   |
> | Welsh | 0.771  | 0.73      | 0.633  | 0.611   | 0.828  | 0.796     | 0.61   | 0.796  |
>
>
> In the revised manuscript, we will also include a related work section that compares our work to other open corpora, as well as complete per-language token stats and benchmark scores in a dedicated appendix chapter.

---

### Author Response · Authors · 2025-12-03
**Rebuttal Revision Comment**

We once again thank all reviewers for their useful suggestions. We have updated the revision, incorporating some of the results achieved during the rebuttal:

* More verbose language distribution accompanied by the tokenizer details and evaluation (now Appendices C and D).

* Per-language benchmark scores in Appendix H.

* Deduplication details in Section 4.

We have also scheduled the final changes for the next revision, using the opportunity to expand the main text to page 10. These changes include the Related Work discussed in the rebuttal and shown in the **General Comment**, as well as different discussion points in a separate section. We will also include more details in the description of the processing tools.

---

### Meta-Review · Area_Chair_QrQ5 · 2025-12-27

**Summary:**

The paper introduces Common Corpus, a 2-trillion-token multilingual dataset comprised entirely of public domain or permissively licensed data, intended to support open and compliant LLM pre-training. The authors also release data curation tools (e.g., OCR correction) and small models trained on this corpus.

The reviewers universally praised the paper for its timeliness, the sheer scale of the open data, and the rigorous documentation regarding provenance and licensing. The transparency of the data pipeline and the focus on "ethical" (permissibly licensed) data were seen as significant contributions to the open-science ecosystem. However, initial concerns were raised regarding the dataset's heavy skew toward European languages despite claims of multilingual diversity, the choice of benchmarks for the trained models, and the lack of direct comparisons to existing open corpora. Some important concerns (e.g., comparisons with existing open corpora) were indeed addressed during the rebuttal.

**Reviewer Concerns:**

**Addressed Concerns:**
* **Comparison to existing corpora:** Reviewers BY7t and GQ3w noted a lack of clear comparison with other open datasets. The authors addressed this in the general comment by providing a table comparing Common Corpus to datasets like Dolma, FineWeb, and KL3M across dimensions of licensing and domain diversity.
* **Benchmarking and Evaluation:** Reviewers aTGr and GQ3w felt the empirical evaluation was limited. The authors responded by providing per-language breakdown scores on MultiBLIMP and justifying their choice of small models and specific benchmarks suitable for base models without instruction tuning.
* **Deduplication and Quality:** Reviewer aTGr questioned the deduplication process. The authors clarified that the institutional nature of the source data minimizes duplication, and specific deduplication measures were taken for the code component (The Stack).

**Outstanding Concerns:**
* **Euro-centricity:** Reviewer aTGr pointed out that the "multilingual" claim is overstated given the Euro-centric nature of the data. While the authors acknowledged this limitation—attributing it to their familiarity with European copyright law—and argued that the dataset is still more diverse than competitors, the heavy skew toward English and European languages remains an inherent property of the release.
* **Terminology:** Reviewer LPfF raised a concern regarding the use of the term "Ethical data," suggesting "legal" or "consensual" might be more accurate. Although the authors argued that "ethical" encompasses broader curation practices beyond legality, the reviewer maintained that the term might not be the precise choice for a scientific paper, though they understood the authors' rationale.

**Reviewer Scores:**

* **Reviewer aTGr:** The score (8) is likely to remain unchanged. The reviewer was already strong in their support, and the authors adequately answered the questions regarding deduplication and language distribution.
* **Reviewer BY7t:** The score (8) is likely to remain unchanged. The rebuttal provided the requested comparison table and clarification on overlap with other open corpora.
* **Reviewer LPfF:** The score (6) is likely to remain unchanged. The reviewer acknowledged the response, but retained reservations about the terminology.
* **Reviewer GQ3w:** The score (6) is likely to move to 8. The authors addressed the scalability and quality assurance questions effectively, though the empirical limitations of training only small models persist.

---

### Decision · Program_Chairs · 2026-01-26

Accept (Oral)